# SARS-CoV-2 mucosal antibody development and persistence and their relation to viral load and COVID-19 symptoms

Janeri Fröberg [1,2,8], Joshua Gillard[1,2,3,8], Ria Philipsen[1,2,4], Kjerstin Lanke[2], Joyce Rust[1,2,4], Diana van Tuijl[1,2,4], Karina Teelen[2], Teun Bousema[2], Elles Simonetti[1,2], Christa E. van der Gaast-de Jongh[1,2], Mariska Bos[1,2], Frank J. van Kuppeveld [5], Berend-Jan Bosch [5], Marrigje Nabuurs-Franssen[6], Nannet van der Geest-Blankert[7], Charlotte van Daal[7], Martijn A. Huynen[3], Marien I. de Jonge[1,2] & Dimitri A. Diavatopoulos [1,2 ✉]

Although serological studies have shown that antibodies against SARS-CoV-2 play an important role in protection against (re)infection, the dynamics of mucosal antibodies during primary infection and their potential impact on viral load and the resolution of disease symptoms remain unclear. During the first pandemic wave, we assessed the longitudinal nasal antibody response in index cases with mild COVID-19 and their household contacts. Nasal and serum antibody responses were analysed for up to nine months. Higher nasal receptor binding domain and spike protein-specific antibody levels at study inclusion were associated with lower viral load. Older age was correlated with more frequent COVID-19 related symptoms. Receptor binding domain and spike protein-specific mucosal antibodies were associated with the resolution of systemic, but not respiratory symptoms. Finally, receptor binding domain and spike protein-specific mucosal antibodies remained elevated up to nine months after symptom onset.

[1] Radboud University Medical Centre, Radboud Institute for Molecular Life Sciences, Laboratory of Medical Immunology, Section Paediatric Infectious Diseases, 6525 GA Nijmegen, The Netherlands. [2] Radboud Center for Infectious Diseases, Radboudumc, Nijmegen, The Netherlands. [3] Centre for molecular and Biomolecular Informatics, Radboud Institute for Molecular Life Sciences, Radboud University Medical Centre Nijmegen, 6525 GA Nijmegen, The Netherlands. [4] RTC CS Radboud Technology Center Clinical Studies, Radboudumc, Nijmegen, The Netherlands. [5] Utrecht University, Faculty of Veterinary Medicine, Department of Biomolecular Health Sciences, Division Infectious Diseases and Immunology, Utrecht, The Netherlands. [6] Department of Medical Microbiology and Infectious Diseases, Canisius Wilhelmina Hospital, Nijmegen, The Netherlands. [7] Department of Occupational Health, Radboud University Medical Centre, Nijmegen, The Netherlands. [8] These authors contributed equally: Janeri Fröberg, Joshua Gillard. ✉email: dimitri.diavatopoulos@radboudumc.nl

The rapid spread of SARS-CoV-2 in populations is attributed to several aspects, i.e. the route of transmission via respiratory droplets, rapid viral replication and shedding from the upper respiratory tract[1], early infectiousness with a peak viral load before the onset of symptoms[2,3] and a high frequency of mild and asymptomatic infections[3–7]. These aspects have complicated effective control of SARS-CoV-2 spread, as containment strategies have primarily been dependent on symptomatic case detection[8,9]. Indeed, pre-symptomatic carriers are likely an important driver of community-based viral transmission[8,10]. Transmission within households contributes significantly to the spread of SARS-CoV-2 in communities, as close contact within households facilitates early-onset transmission of the virus[11–13].

Antibodies are considered to play a crucial role in protection against viral (re)infection. The SARS-CoV-2 virus enters human cells following binding to the ACE2 receptor with the receptor-binding domain (RBD) of the viral spike (S) protein. Serological studies have shown that antibodies directed against the spike protein and RBD region are capable of neutralising viral binding and entry, and vaccines inducing immunity against the S-protein have been shown to be efficacious[1,3,14–17]. Infection with SARS-CoV-2 also induces humoral responses against the viral nucleocapsid (N) protein. The N-protein of SARS-CoV-2 shares~80% of its amino acid sequence with SARS-CoV-1 and other seasonal coronaviruses[18]. Therefore, pre-existing immunity against the N-protein may play a protective role during infection[18,19]. Studies investigating antibody response dynamics in mild cases have demonstrated the development of serum antibodies against SARS-CoV-2 ~10 – 15 days post symptom onset[1,20]. An understudied aspect of the immune response to SARS-CoV-2 infection is the magnitude, kinetics and persistence of mucosal antibodies. Animal studies of other human coronaviruses have shown that mucosal antibodies play a key role in the reduction of viral load and may contribute to protection following re-exposure. Intranasal vaccination also induces strong and protective mucosal immune responses[21,22]. Viral entry and replication of SARS-CoV-2 first occur in the upper respiratory tract, where ACE2 receptor expression is very high[23,24]. Together, these findings suggest that nasal antibodies may play a key role early in the infection. The composition of mucosal antibodies differs from serum, particularly with regards to secretory IgA (sIgA) and IgM (sIgM). sIgA is primarily dimeric, whereas serum IgA is predominately monomeric, which may affect both viral neutralisation and the inflammatory response[25–27].

To obtain a comprehensive view on the development and persistence of mucosal antibodies following mild SARS-CoV-2 infection, we performed a prospective, observational household-contact study in 50 households with at least one PCR-confirmed case (index case) and two household members (contacts). We assessed the timing, magnitude and persistence of mucosal antibodies against SARS-CoV-2 antigens and examined their associations with viral load and COVID-19 related symptom development.

Based on PCR positivity and/or seropositivity, household contacts were classified into cases and non-cases. Baseline mucosal antibody levels were associated with variation in viral load. The development of COVID-19 symptoms during the 28-day follow-up period was analysed in relation to mucosal antibody dynamics. Our study provides new insights in mucosal antibody production during a primary mild SARS-CoV-2 infection and the longevity of antibodies in nasal fluid and serum.

## Results

**Cohort description and study design.** The recruitment strategy for inclusion of households focused on healthcare workers with a PCR-confirmed infection who were in home isolation (index cases), with at least two participating household members. Between 26 March 2020 and 15 April 2020, i.e. during the first pandemic wave, we prospectively enrolled 50 index cases and 137 household members (Fig. 1a, b). Index cases were mostly female (76%), reflective of the gender distribution amongst healthcare workers, with a median age of 46 (IQR: 37–54). Consequently, household contacts were mostly male (61%) and younger, with a median age of 21 (IQR:13-46) (Supplementary Table S1). An overview of the study design is shown in Fig. 1c. Home visits were performed to collect naso- and oropharyngeal swab samples and nasal mucosal lining fluids (MLF) at study start (D0). Study participants self-sampled MLF on three subsequent timepoints as described in 'Methods', and a serum sample was collected via fingerprick on day 28. Index cases were asked to report their first day of symptoms, and all participants completed a daily symptom survey during the 28-day follow-up to monitor symptom development (Fig. 1c). Contacts were classified as cases or non-cases based on PCR and/or seropositivity (see 'Methods'). To analyse the persistence of nasal and serum antibodies, serum ($n = 100$) and MLF ($n = 108$) was collected from index and contact cases at nine months after study enrolment (Fig. 1a, c).

**High infection rate among household contacts.** All participants were tested for SARS-CoV-2 infection at study day 0 by PCR on naso- and oropharyngeal swabs. Antibody levels in serum and MLF were measured using a fluorescent-bead-based multiplex immunoassay (MIA). IgG, IgA and IgM levels specific for S-, N- and RBD antigens were determined. To determine increases in antibody levels, we used 32 pre-pandemic serum control samples and 17 pre-pandemic MLF control samples. Antibody values were normalised by calculating the Log2-transformed antibody levels of study samples over the mean of the control samples. To evaluate the performance of the MIA, we performed a ROC analysis with the pre-pandemic controls as a negative control and the PCR-positive samples as a positive control. For the RBD and S antigen, the MIA performed well for all antibodies in both MLF (AUC > 0.750) and serum (AUC > 0.915), with the S-protein analysis in serum performing the best of the tested antigens (AUC > 0.970, Supplementary Fig. S1a). The MIA assay showed high reproducibility when two batches analysed 9 months apart were compared, with Spearman correlations >0.87 for all antibody/antigen combinations (Supplementary Fig. S1b). Based on the ROC results, serum anti-SIgA, IgM and IgG antibody responses on day 28 were selected as a measure to identify cases, in combination with the PCR analysis performed at study start.

Based on PCR positivity on day 0 and/or seropositivity against the S antigen on day 28, we identified 80 contact cases among the 137 household contacts (58.4%, Supplementary Table S1). To examine potential age-related differences, we stratified household contacts into three age categories, i.e. <18 years ($n = 46$), 18–49 years ($n = 54$) and ≥50 years ($n = 29$). No age-related differences were observed with regards to the frequency of cases amongst household contacts, i.e. 57% for <18 years, 58% for 18–49 years and 61% for ≥50 years. Of note, a large percentage of contacts was already PCR positive at the study start, especially in the ≥50 years age group (45.2%, see Fig. 2a). Almost all PCR + contact cases were seropositive (see Supplementary Table S2). We also examined 'nasoconversion' against the S-protein, which shows that 68% of the household contacts had become 'naso-positive' by day 28. Seropositivity at day 28 identified 32 additional cases among the PCR- household contacts (40%), of which 11 had also nasoconverted (34%, Supplementary Table S2). Highest seroconversion among household contacts was observed for anti-S IgG and IgA, with lower

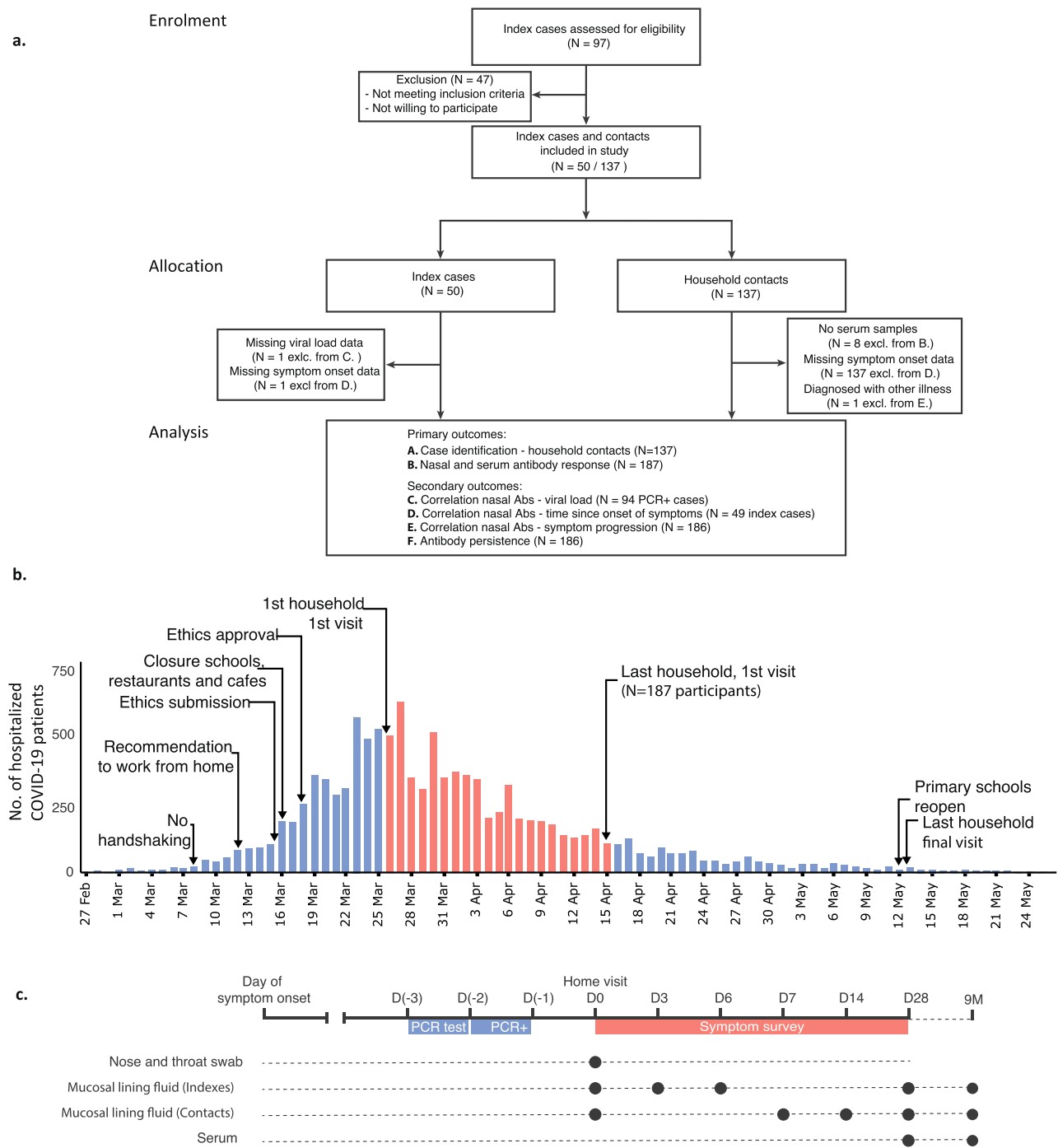

**Fig. 1 Flow diagram and study procedures. a** Flow diagram describing the recruitment of households, sample sizes, and study outcomes. We initially contacted 97 index cases that were tested positive for SARS-CoV-2. After the exclusion of cases that did not meet the inclusion criteria or did not consent, 50 index cases and their household contacts ($n = 137$) were recruited. Mucosal lining fluid (MLF) antibodies were analysed as a primary outcome in both indexes and household contacts. Secondary analyses (correlation of MLF antibodies with viral load, time of symptom onset and symptom progression and antibody persistence) were performed. **b** Study timeline, with respect to the number of hospitalisations due to COVID-19 over time and COVID-19 control measures in the Netherlands[46,47]. The first home visit was conducted at the peak of hospitalisations at March 26, and the last visit was one day after the reopening of primary schools, at May 13. **c** Overview of the study design and measurements. Home visits were initiated after the index was tested positive for SARS-CoV-2 by PCR, to collect naso- and oropharyngeal swabs for viral load determination as well as nasal MLF samples. Subsequent MLF samples were collected and stored by the participants, who also completed a daily symptom survey. At the end of the 28-day follow-up, blood samples were collected for serological analyses. A subset of cases ($n = 108$) was visited again 9 months after enrolment. At this timepoint, a MLF and serum sample were taken. Source data are provided as a Source Data file.

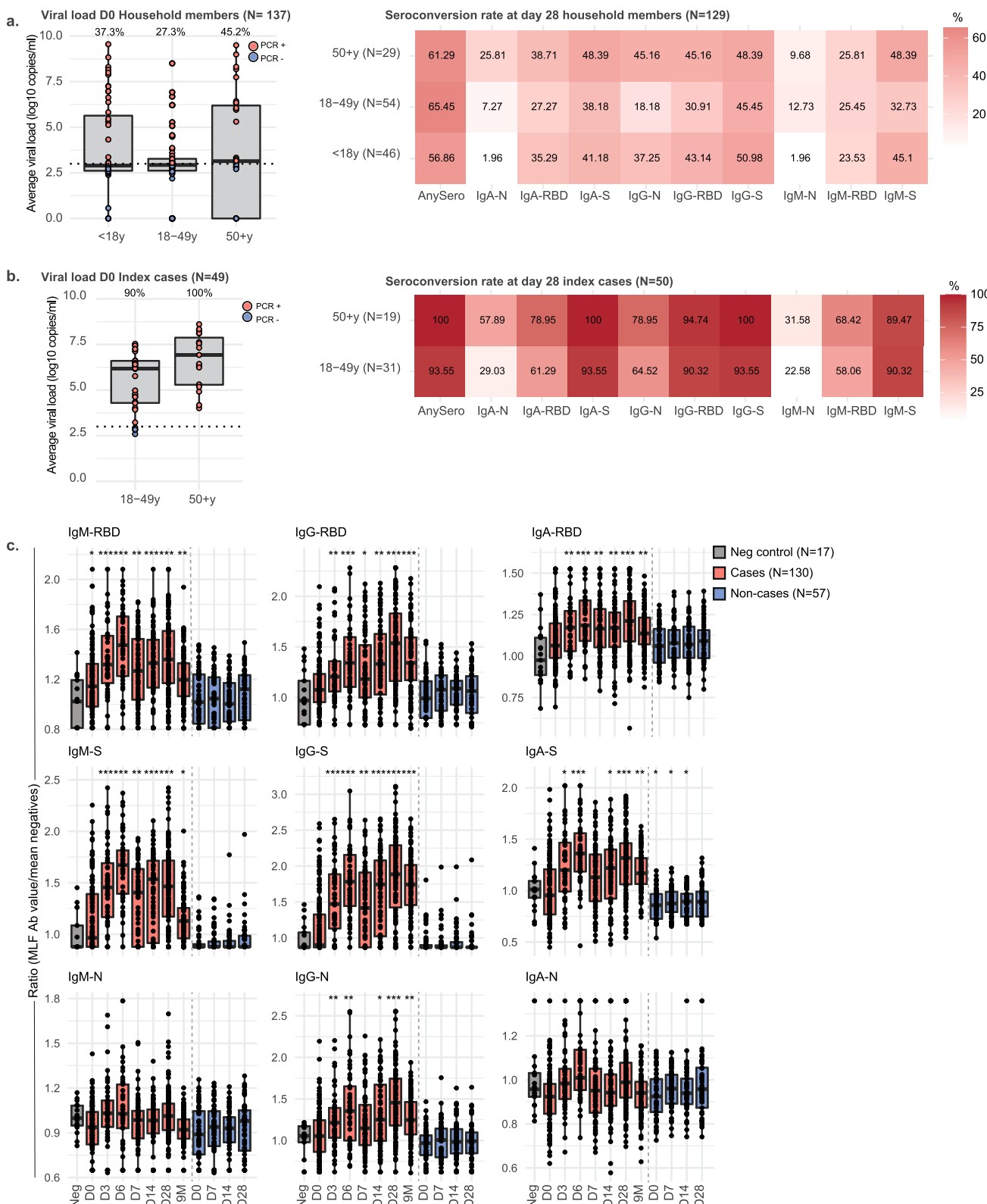

seroconversion against N (Fig. 2a). Of the index cases, 94% was still PCR positive at study start (Fig. 2b). Of the index cases, 92% had 'nasoconverted' by day 28, most of whom were PCR + and sero + (Supplementary Table S2). Similar to the contact cases, seropositivity rates were highest for S and lowest for N. IgG and IgA seropositivity rates were generally higher than IgM (Fig. 2b).

The magnitude of serum antibody levels in the index and contact cases did not differ between the three age groups, except for anti-N IgA and IgG, which were significantly higher in the ≥50 years old (Supplementary Fig. S2a). To examine the correlation between nasal and serum antibody levels in cases, we focused on the day-28 and 9-month timepoints. For all

**Fig. 2 Mucosal antibodies against Spike increase after SARS-CoV-2 infection. a** PCR and seropositivity rate of household members ($n = 137$), split into age groups. PCR positivity was defined as a Ct value <36, which corresponds to a viral load of at least $10^3$ copies/ml extracted sample (dashed line). The boxplots display a median line, interquartile range (IQR) boxes, 1.5*IQR whiskers and individual data points (red=PCR positive, blue=PCR negative). The percentage of PCR-positive individuals is depicted in the figure. Seroconversion was defined as an antibody titre that is higher than the mean + 2*standard deviation of the pre-pandemic control samples and is given for each antigen and antibody isotype measurement, as well as for any single antibody measurement (anySero). **b** PCR positivity and seroconversion rates of index cases ($n = 49$), split into age groups. **c** Mucosal antibody responses to RBD, S and N, plotted as a ratio to the pre-pandemic negative controls ($n = 17$), per study timepoint and split into cases ($n = 130$, red boxplots) and non-cases ($n = 57$, blue boxplots). Cases were defined with the PCR positivity on day 0 and seropositivity against the Spike protein on day 28. The boxplots display a median line, interquartile range (IQR) boxes, 1.5*IQR whiskers and individual data points. Pre-pandemic controls are presented in the grey boxplots for comparison ($n = 17$). Values of each boxplot were compared with the negative controls with the two-sided Wilcoxon signed-rank test, *$P < 0.05$, **$P < 0.01$, ***$P < 0.001$. There were no non-cases in the D3, D6 and 9 M timepoint. Source data are provided as a Source Data file.

antibody–antigen combinations, positive and significant correlations were observed between MLF and serum (Supplementary Fig. S2b).

**Nasal antibodies against SARS-CoV-2 increase after infection and persist up to 9 months**. Next, we investigated nasal antibody response dynamics (Fig. 2c). On day 0 and day 28, MLF samples were collected from both index cases as well as from all household contacts. MLF sampling timepoints for the index cases were chosen to capture the antibody response during the acute phase of infection, i.e. days 3 and 6. Because the infection status of the household contacts was not yet known at the moment of inclusion, MLF sampling timepoints from household contacts were selected to capture possible secondary infections, i.e. days 7 and 14. To examine antibody persistence, additional MLF and serum samples were collected from identified cases at 9 months.

By day 3 of the study, nasoconversion had already occurred for all antibody measurements, except for anti-N IgA and IgM. Although waning was observed between 28 days and nine months, all antibody isotypes against S and RBD as well as anti-N IgG remained significantly elevated (Supplementary Fig. S3). Thus, primary infection with SARS-CoV-2 induces a broad and persistent mucosal antibody response against Spike and RBD, whereas for nucleocapsid protein the response was restricted to IgG.

**Early nasal antibody production is correlated with lower viral load**. Since participants were included at various stages of infection, there is significant variation in the viral load and mucosal antibody levels at the onset of the study, i.e. day 0. To examine potential relationships between antibody levels and viral load, we focused on day 0 for which we had paired viral load and mucosal antibody measurements. Correlation analysis indicated viral load was negatively correlated with anti-S and RBD mucosal antibodies, of which IgM-S showed the strongest correlation (Fig. 3a and Supplementary Fig. S4a). Of note, participants with a longer interval between onset of disease and inclusion into the study showed higher mucosal IgA levels (Fig. 3a and Supplementary Fig. S4b).

Previous studies have shown that SARS-CoV-2-specific antibodies become detectable in serum at approximately 2 weeks PSO[28–30]. Here, we assessed the relationship between longitudinal mucosal antibody responses and symptom onset in more detail. This analysis was focused on index cases only, since information on the exact days post symptom onset (PSO) was available for this group only. Longitudinal nasal antibody responses were assessed by binning samples into 3-day timeframes relative to the day of symptom onset and plotting the values alongside controls (Fig. 3b). Mucosal IgM, IgA and IgG antibody levels for S and RBD antigens were significantly elevated relative to controls between 7 and 9 days PSO, while for the N-protein only IgG antibody responses were significantly higher than controls.

Although IgA-N showed a similar response pattern, this did not reach statistical significance, presumably due to high variation in the pre-pandemic control samples. Nasal antibody responses that were increased after infection remained significantly elevated up to 9 months PSO.

**Increases in nasal antibodies against S and RBD are associated with the resolution of clinical symptoms**. Next, we explored potential relationships between nasal antibodies and the progression of COVID-19 symptoms. Since none of the participants in our study required hospitalisation or other medical intervention, our study population represents a cross-section of mild COVID-19 cases in a community setting. We examined the progression of 23 symptoms using a survey that all volunteers filled in daily throughout the 28-day follow-up. Symptoms were grouped into three categories: gastrointestinal symptoms (GS), systemic disease symptoms (SDS) and respiratory symptoms (RS). The most frequently reported symptoms were respiratory symptoms, which were also frequently reported by non-cases (Supplementary Fig. S5a). Anosmia/dysnosmia, i.e. change or loss of taste and smell, and systemic symptoms including a loss of appetite, muscle pain, joint pain, chills, fatigue and fever were reported significantly more often in cases than in non-cases. We examined whether the symptom duration varied between different symptom categories by generating a Kaplan–Meier curve for each symptom type. GS was excluded from this analysis as it only contained three symptoms. We found that systemic disease symptoms generally resolved faster than respiratory symptoms ($P$ value: 0.02, Supplementary Fig. S5b), with 50% of the cases being SDS free by day 14 after study inclusion. To examine longitudinal changes in the number of COVID-19 symptoms and identify potential differences between different age groups, we binned symptom notifications into 3-day timeframes relative to the study day, similar to the nasal antibody analysis described above. Subsequently, we analysed symptom progression per age group. Although the faster resolution of SDS than RS was observed across all age groups, the number of reported symptoms per 3-day period was lowest in the <18 years group and increased in both older age groups (Fig. 4a). To assess the contribution of age to clinical symptom progression, we constructed a linear mixed-effects model per symptom group with the number of reported symptoms as the response and study day, age, viral load at study start and sex as covariates. Such a linear model was a good fit for our longitudinal symptom data (Supplementary Fig. S6). While time was—as expected—significantly associated with decreases in symptoms, increased age was significantly associated with increased SDS and RS when correcting for the effect of time ($P$ value: 0.0013 and 0.0001, respectively, Fig. 4b). Female sex was associated with marginally more SDS ($P$ value: 0.02, Fig. 4b). Viral load at day 0 was not related to SDS or RS progression ($P$ values: 0.78 and 0.92, respectively). To ascertain whether the induction of nasal antibodies was associated with COVID-19

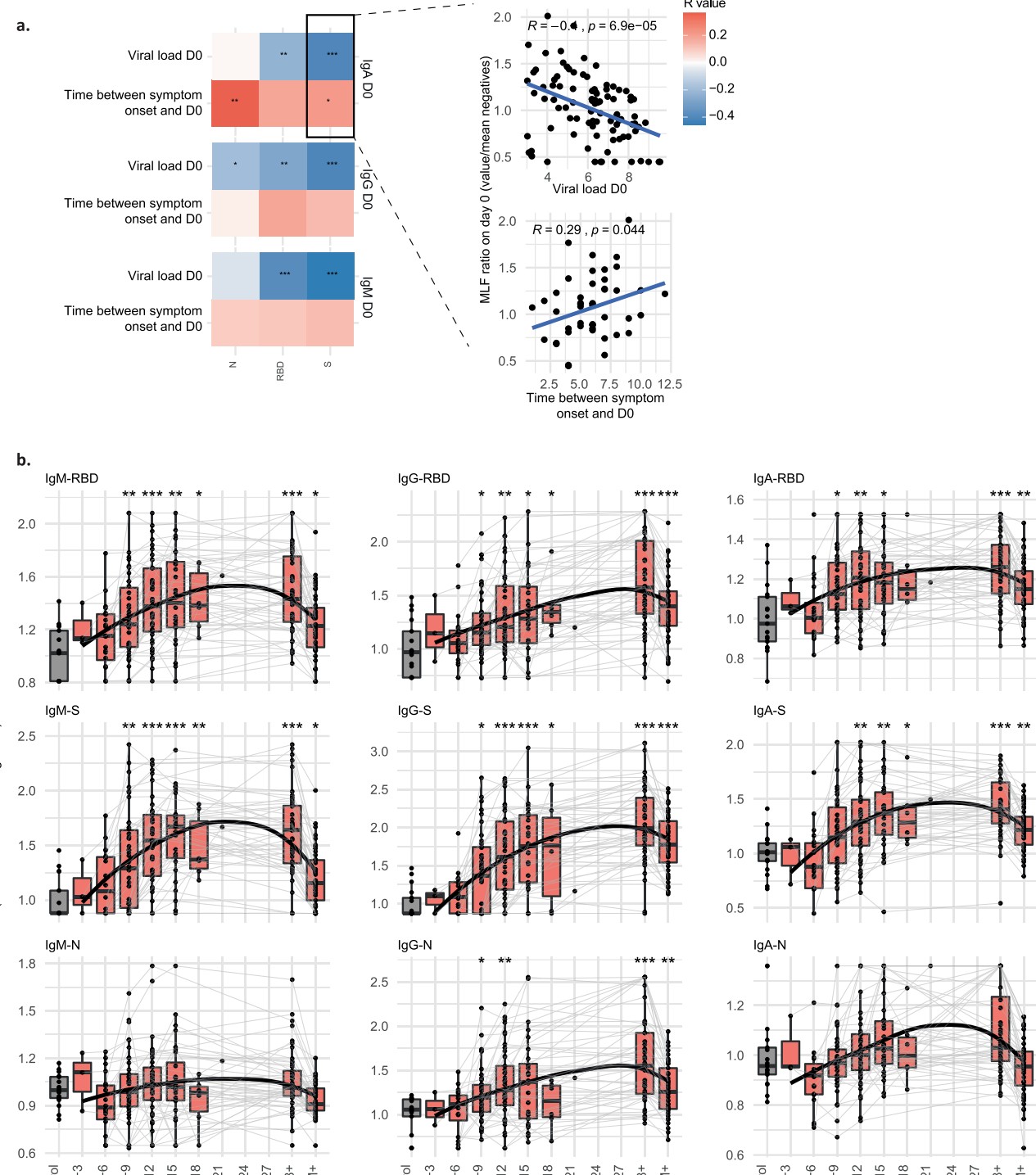

**Fig. 3 Mucosal antibody responses are correlated with viral load and persist up to nine months after symptom onset. a** IgM, IgG and IgA antibody responses against SARS-CoV-2 spike (S), receptor-binding domain (RBD) or nucleocapsid (N) collected in mucosal lining fluid at study start were correlated with viral load at study start and time between symptom onset and study start. Two-sided Spearman correlations were calculated, *$P < 0.05$, **$P < 0.01$, ***$P < 0.001$. The correlation plots of IgA-S are shown as an example, all correlation plots can be found in Supplementary Fig. S4, together with the exact $P$ values of the correlations. **b** Longitudinal mucosal antibody responses to S, RBD and N, plotted as a ratio to mean of the pre-pandemic negative controls ($n = 17$, grey boxplots), relative to the days post symptom onset. The boxplots display a median line, interquartile range (IQR) boxes, 1.5*IQR whiskers and individual data points. Values within each timeframe were compared with the controls with the two-sided Wilcoxon signed-rank test. A non-parametric Loess curve is shown to visualise the trend over time. Measurements from the same individual are connected with a grey line. Source data are provided as a Source Data file.

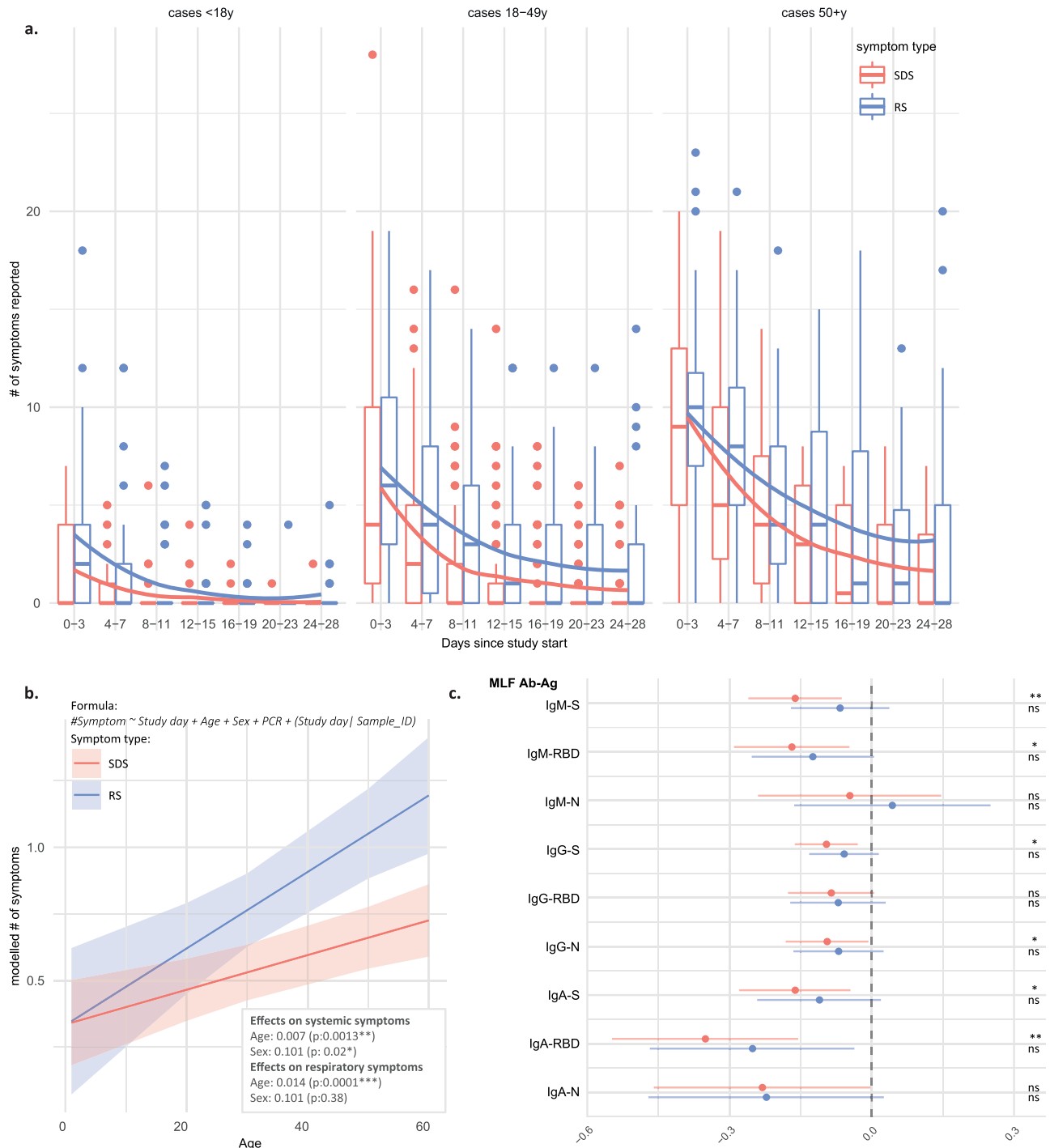

symptoms, changes in nasal antibody levels over time were univariately added to the mixed-effects model. This way, the effect of an increase of antibody signal on the changes in the number of reported symptoms could be assessed while correcting for the effect of time, viral load, age and sex. Overall, increases in all nasal antibodies that were increased after infection were associated with decreased SDS and RS. After correction for multiple testing, a significant association was only found for SDS, where high levels of all isotypes against S, IgA and IgM against RBD and IgG against N were related to a decrease in symptoms. The largest effect was seen in the relation between IgA-RBD and SDS (Fig. 4c). The age effect remained significant for all combinations.

## Discussion

In this study, we examined the development and persistence of nasal antibodies following infection with SARS-CoV-2 in a household-contact study. Out of the 137 household contacts, we identified 80 cases. We found that mucosal antibodies against S and RBD increase 7–9 days after symptom onset and remain elevated for at least 9 months. Anti-S and RBD mucosal antibodies were found to be correlated with a lower viral load and were related to a faster decline in systemic symptoms.

We identified a very high frequency of cases among the household contacts (54.7%), which is higher than previously reported (11–37%)[31–33]. Most household transmission studies

**Fig. 4 Age and mucosal antibody levels influence the presence and reduction of COVID-19-related symptoms. a** The number of respiratory (RS, blue) and systemic disease (SDS, red) symptoms were recorded for all cases (n = 130) for each day during the 28-day study period. Data are plotted relative to the study day and values were binned into 3-day timeframes. The boxplots display a median line, interquartile range (IQR) boxes, 1.5*IQR whiskers and outlier data points. A non-parametric LOESS curve is shown as a red (SDS) or blue (RS) line in order to visualise the trend over time. Cases were grouped into three age groups: <18 years, 18–49 years, and 50 + years. **b** A linear mixed-effect model (MEM) was fit to the data per symptom group. The response was specified as the number of symptoms on a given day, and explanatory fixed effects variables were: Study day, age, sex and viral load at study start (PCR). The study day was also specified as a random slope, and Sample ID as a random intercept. The model fit per individual symptom trajectory can be found in Supplementary Fig. S6. A significant effect of age was demonstrated for RS and SDS, as well as an effect of female sex on SDS. The mean predicted symptom values are plotted against age with 95% confidence interval bands (based on the standard error of the mean), and model estimates and p-values are depicted in the figure, together with the used model formula. **c** Scaled longitudinal mucosal antibody measures were univariately added to the already existing MEM formula depicted in panel **b**. The predicted change in symptoms per unit increase of the scaled relative antibody level is presented with 95% confidence intervals based on the standard error of the mean, and two-sided P values for the association are plotted on the right, corrected for multiple testing with the Benjamini & Hochberg method (ns P > 0.05, *P = 0.01 for IgG-S, IgA-S and IgM-RBD, and P = 0.046 for IgG-N, **P = 0.004 for IgM-S and IgA-RBD on SDS. None of the antibody effects on RS were significant). Source data are provided as a Source Data file.

conducted during the first wave identified cases based on a single PCR test. This likely underestimates the true number of cases within a cluster, as PCR positivity is dependent on the time of sampling in relation to the infection. By combining PCR with seropositivity at 28 days after study inclusion, we were able to identify an additional 32 cases compared to PCR testing alone, increasing the total number of contact cases by 40%. This underlines the importance of antibody testing to assess disease exposure and transmission, especially in settings where PCR testing is limited and/or rapid antigen tests are not available.

We assessed the antibody production in the nose at multiple timepoints and observed a significant increase in nasal antibody levels in response to infection, with similar kinetics as previously described for serum and saliva[3,29,30]. The majority of sero- and PCR-positive contacts were also naso-positive by day 28 (Supplementary Table S2). Nasal antibodies correlated strongly with serum antibodies, although correlations were noticeably weaker for IgA, which has previously also been observed in saliva[29,30,34]. As expected, both serum and nasal antibody levels decreased over time but remained above detection threshold for at least 9 months. These results substantiate the possibility of using nasal fluid as an epidemiological tool to monitor disease exposure until long after infection.

An effective early antibody response can modulate the clinical course of infection, as observed with e.g. influenza virus and Chikungunya infections[35,36]. We, therefore, examined the relationship between the early nasal antibody response following SARS-CoV-2 infection and the progression of symptoms over time. Our study found that most anti-SARS-CoV-2 nasal antibodies, but in particular anti-RBD IgA correlated with the resolution of systemic disease symptoms. Surprisingly, no relation was identified with the resolution of respiratory symptoms. It should be noted that the trajectory of respiratory symptoms was different compared to systemic symptoms. Furthermore, respiratory symptoms were reported much more frequently, including in non-cases, and were less specific to COVID-19 (Supplementary Fig. S5a). Although the exact nature of the relationship between nasal antibodies and clinical symptoms requires further investigation, higher anti-spike/RBD nasal antibodies at the study baseline did correlate with lower viral load, particularly for anti-RBD/S IgM. Possibly, early control of viral infection in the upper respiratory tract reduces the shedding of virus and viral replication in the lower respiratory tract and the periphery, resulting in less systemic symptoms. These findings are in line with a study by Butler et al., who also noted a prominent role for salivary anti-RBD IgA[37] neutralising antibodies in reducing clinical severity.

The nucleocapsid is highly immunogenic and abundant in coronaviruses and is conserved across both SARS-CoV-2 and pre-COVID-19 seasonal coronaviruses. Although IgG and IgA

antibodies against the nucleocapsid increased following infection, anti-N IgM levels did not increase in cases in our study. These results suggest the response to N is a memory recall response rather than a primary response, although variation between patients existed and some cases did show an anti-N IgM response. We used pre-COVID-19 MLF samples to determine 'naso-conversion'. Although this exploratory approach worked well for S/RBD for which no pre-existing antibodies are present, the presence of pre-existing anti-N nasal antibodies makes it more difficult to establish a valid 'cut-off' level.

Our study has several limitations. First, the starting point was the inclusion of healthcare workers, most of whom were female, and thus not entirely representative for the larger population. It should be noted that this study was performed during the first wave in March–April of 2020, when all schools in the Netherlands were closed (Fig. 1b) and therefore no conclusions can be drawn in relation to child–child contacts in the transmission of SARS-CoV-2. When we performed the study, we were unable to collect additional samples for viral PCR due to national shortages in swabs and transport medium, limiting our ability to further examine the dynamic interactions between viral infection and antibody responses. To minimise the study burden, we made an initial decision to limit the number of serum samples to the day 28 timepoint, to which we later added the nine months timepoint. Consequently, no comparison of serum and nasal antibody levels is possible at the early timepoints before day 28 of the study. Nonetheless, the possibility for participants to self-sample nasal MLF repetitively in a non-invasive manner removes an important obstacle for use in age groups that are normally difficult to sample, such as children, or hard-to-reach locations.

Taken together, our study shows that an early and higher nasal antibody response may play a key role in limiting disease by initiating early viral clearance and facilitating the resolution of systemic symptoms. Further research is needed to validate the role of nasal antibodies in clinical protection. Nasal IgA and IgG antibodies remain detectable for at least nine months after infection and likely confer at least partial protection against re-infection. Since mucosal antibodies are the first line of defence against viral infection, monitoring post-infection and post-vaccination nasal antibody levels may allow us to identify early signs of the waning immunity against infection. Finally, the study design and analysis strategy presented here can be used as a blueprint for follow-up investigations not only for COVID-19 but also for other infectious diseases.

## Methods

**Recruitment**. This observational prospective cohort study was conducted among COVID-19 cases with a laboratory-confirmed infection, as well as their household members that remained in home quarantine at the same address. The study was

conducted in accordance with the provisions of the Declaration of Helsinki (1996) and the International Conference on Harmonisation Guidelines for Good Clinical Practice. The study was approved by the local medical research ethics committee (CMO Regio Arnhem-Nijmegen) and is registered with ClinicalTrials.gov (NCT04590352; ethical committee reference NL73418.091.20). All index cases in this study were healthcare workers (HCW) from three hospitals (Radboudumc, CWZ and Rijnstate) in the provinces of Gelderland in the Netherlands, with a confirmed SARS-CoV-2 infection. Study participants were included from March 26, 2020 until April 15, when the inclusion number of 50 households was reached. Participants were introduced to the study through the occupational health and safety services (OHS) of the participating hospitals. HCWs were included if they had a positive Polymerase Chain Reaction test (PCR test) for the SARS-CoV-2 virus, tested and judged by the OHS of their hospital, with a positive indication for home isolation, and had at least two household members willing to participate.

**Study design.** Before the first home visit, all index cases of the family had a telephone interview, where they were asked about their first day of symptom onset, whether they were in isolation from the rest of the household, whether physical contact was restricted with other household members, whether they were still symptomatic, and whether they thought they were the primary case in the household. Households were visited within 1–2 days of a positive PCR in the index case. Following informed consent, nasopharyngeal and oropharyngeal swabs were taken for viral PCR, as per diagnostic guidelines[38]. A nasal mucosal lining fluid (MLF) sample was obtained from all participants by the use of the Nasosorption™ FX·i nasal sampling device (Hunt Developments, UK). A synthetic absorptive matrix (SAM) strip was gently inserted into the nostril of the participant and placed along the surface of the inferior turbinate. The index finger was lightly pressed against the side of the nostril to keep the SAM strip in place and to allow MLF absorption for 60 s, after which the SAM strip was placed back in the protective plastic tube. Participants were instructed on how to self-sample MLF at home. Finally, participants were asked about their symptoms of that day.

Participants were followed up for ~28 days, starting on the day of the first home visit (day 0) and ending on the last home visit (days 28–33). This range in the last visit was due to logistical difficulties during the summer holidays; 14 index cases were visited on their day 29, three on day 30, five on day 31, four on day 32 and one on day 33. All participants registered their symptoms for 28 days. During follow-up, clinical symptoms were registered three times daily and MLF was collected at three different study days via self-sampling. For the index case, MLF was collected on days 0, 3, 6 and one of the days 28–33 and for the household contacts this was on days 0, 7, 14 and one of the days 28–33 (Fig. 1c). Self-sampled MLF samples were stored in biosafety bags in the participants' own freezer (temperature around −20 °C).

At the final home visit, MLF samples were picked up and transported to the Radboudumc on dry ice, where they were stored at −80 °C until further testing. For antibody analysis, Nasosorption™ FX·i nasal sampling devices were thawed on ice, after which the synthetic adsorptive matrix (SAM) was removed using sterile forceps. The SAM was placed in a spin-X filter Eppendorf tube with 300 μl of elution buffer (PBS/1% BSA) for a minimum of 10 min, followed by centrifugation at 16,000 × g for 10 min at 4 °C. To prevent unspecific binding, the spin-X filter columns were pre-incubated with the elution buffer for 30 min. The filter cups were then removed from the Eppendorf tubes using sterile forceps. To inactivate live SARS-CoV-2, the eluate was incubated for a minimum of 45 min at 56 °C, spun down, aliquoted and stored at −20 °C for further testing.

Finger-prick blood (~0.3 ml) was collected from all participants consenting to the fingerprick at day 28 by the use of a sterile disposable lancet device (BD Microtainer Lancet) and a sterile capillary tube. Blood samples were kept at room temperature until processing at the Radboudumc laboratory site, after which serum was stored at −20 °C until further testing.

All collected symptom diaries were digitalised into Castor EDC, clinical trial software for electronic data capture and clinical data management.

Nine months after the first visit, cases ($n = 108$) were visited again and a serum and MLF sample were taken and processed in the manner described above (Fig. 1c). Nobody had been vaccinated yet at that time.

### Sample analysis
*Detection of SARS-CoV-2.* The presence or absence of SARS-CoV-2 and viral copy number per ml was determined on the combined nasopharyngeal and oropharyngeal swab, using a PCR protocol that was developed at the National Institute of Health and the Environment (RIVM), and has been widely used for the diagnosis of SARS-CoV-2 in the Netherlands[39]. The protocol was slightly adjusted for the use of a different reaction mix by the Medical Microbiology Laboratory of the Radboudumc. In short, nasopharyngeal and oropharyngeal swabs were collected in GLY medium and stored at −80 °C until processing. Samples were thawed, vortexed and 500 μl of the sample was lysed in 450 μl MagNAPure Lysis/binding buffer (Roche). An ivRNA internal extraction control was added and samples were extracted on the automated MagNAPure LC 2.0 system using the MagNAPure LC Total Nucleic Acid Isolation kit—High Performance (Roche). Samples were eluted in 50 μl of which 5 μl was used in the RT-qPCR using the Luna Universal Probe One-Step RT-qPCR kit (NEB) with 400 nM E-gene primers (FW: 5'- acaggtacgt-taatagttaatagcgt-3' RV: 5'-atattgcagcagtacgcacaca-3') and 200 nM E-gene probe

(5'-FAM- ACACTAGCCATCCTTACTGCGCTTCG-BHQ1-3' (Biolegio)) on a CFX96 Real-Time PCR Detection System (Bio-Rad). See Supplementary Table S3 for a summary of the primers used. Transcript quantities were calculated using a tenfold dilution series of E-gene ivRNA. The extraction efficiency was checked in a separate RT-qPCR using the Luna Universal Probe One-Step RT-qPCR kit (NEB) with primers targeting the ivRNA that was added prior to extraction.

*Antibody measurements.* For antibody analysis, a fluorescent-bead-based multiplex immunoassay (MIA) was developed. The stabilised pre-fusion conformation of the ectodomain of the Spike protein (amino acids 1–1213) fused with the trimerization motif GCN4 (S-protein) and the receptor-binding domain of the S-protein (RBD) as previously described by Wang et al.[40], and the Nucleocapsid-His recombinant Protein (N) (40588-V08B, Sino Biologicals), were each coupled to beads or microspheres with distinct fluorescence excitation and emission spectra, essentially as described in the paper by den Hartog et al.[41]

A total of six reference serum samples were selected from PCR-confirmed COVID-19 patients with varying immunoglobulin G (IgG) concentrations, and pooled to create standard curves for IgG, IgA and IgM. Next to this, four different samples from the same cohort were used as quality control samples. As negative control samples, we used historical serum ($n = 32$) and MLF ($n = 17$) samples collected prior to the SARS-CoV-2 pandemic.

MLF samples were diluted 1:5 in assay buffer (PBS/1% BSA/0.05% tween-20) and serum samples were diluted 1:500 in assay buffer, incubated with antigen-coated microspheres for 30 min at room temperature while shaking at 450 rpm. Following incubation, the microspheres were washed two times with PBS, incubated with phycoerythrin-conjugated goat anti-human, IgG (Jackson Immunoresearch, 109-116-170), IgM (Southern Biotech, 2022-09) and IgA (Southern Biotech, 2052-09) for 20 min and washed twice. Data were acquired on the Luminex FlexMap3D System. Validation of the detection antibodies was obtained from a recent publication using the same antibodies and the same assay[41], and specificity was checked using rabbit anti-SARS SIA-ST serum.

S- and N-coupled microspheres were combined to measure antibodies directed against multiple antigens (or epitopes) in one single sample. Since antibodies against the S-protein and RBD may compete for the same epitopes, antibody binding to RBD was measured separately. Using different conjugates, IgG, IgA and IgM-specific antibodies concentrations were measured in MLF and serum.

MFI was converted to arbitrary units (AU/ml) by interpolation from a log-5PL-parameter logistic standard curve and log–log axis transformation, using Bioplex Manager 6.2 (Bio-Rad Laboratories) software and exported to R-studio. Negative control samples (MLF and serum) were used to filter out background signals in the antibody measures. The MLF samples originated from the KIRA study performed at the Radboudumc, in which healthy healthcare workers are vaccinated against pertussis as per routine care, and gave consent to the use of the MLF samples for other research. The serum samples originated from the Radboudumc Biobank, which allows the use of serum samples for research as long as the privacy of the donors is guaranteed. The standard dilution range plus four quality control samples were added to each plate. A ROC analysis was performed to analyse the performance of the MIA (Supplementary Fig. S1a). During the analysis of the samples taken nine months after study start, an aliquot of the day 28 samples was thawed and re-analysed, to ensure reproducibility of the assay and enable batch correction if needed (Supplementary Fig. S1b).

**Symptom categorisation.** To analyse the relation between symptom clearance in index cases and the mucosal antibody response, we categorised our set of symptoms into three categories, based on their clinical presentation. This resulted in a set of 23 symptoms, which were categorised into three categories, i.e. respiratory symptoms (RS) systemic disease symptoms (SDS), and gastrointestinal symptoms (GS). RS includes chest pain, sneezing, nose bleeding, pain when breathing, coughing with mucus, dyspnoea, sore throat, loss or change of taste/smell (dysnosmia), coughing and rhinorrhoea. SDS includes dizziness, headache, fever, temperature, chills, joint pain, muscle pain, swollen lymph nodes, low appetite and fatigue. GS includes vomiting, diarrhoea and nausea.

**Case definition.** For analysis of SARS-CoV-2 exposure within households, we categorised the household contacts into cases and non-cases. Cases were defined as being either PCR positive at study start and/or seropositive for IgA, IgG or IgM against S at study day 28. PCR positivity was set on a Ct value<36, which corresponds to a viral load of at least $10^3$ copies/ml extracted sample. The seroconversion threshold was based on the mean + 2*SD of the historic negative control samples, which were collected before SARS-CoV-2 was introduced in the Netherlands. For explorative analyses, a nasoconversion threshold was based on the mean + 2*SD of the historic negative control samples of the MLF samples, and a participant was called "nasoconverted" at day 28 when they had at least one antibody isotype targeted against S above this threshold.

**Statistical analyses and reproducibility.** Analysis of Luminex data was performed with Bio-Plex 200 in combination with Bio-Plex Manager software (Bio-Rad Laboratories, Hercules, CA). All MIA experiments were performed once, but standard reference curves and quality control samples were identical throughout the experiment, to control for possible batch control. Demographical data was

exported from Castor EDC, and double-checked with the paper records by two members of the research team. All statistical analyses were performed using the R-studio environment, with libraries 'stats' (hypothesis tests and correlations), "lme4"[42], "lmerTest"[43] for mixed-effects modelling and associated $P$ values, "ROCit" for the ROC analysis of the MIA and "survival"[44] for Kaplan–Meier survival analysis. The libraries "survminer", "patchwork" and "ggplot2" were used for visualisation. Changes in serum or mucosal antibodies compared to negative controls were tested using a two-tailed paired Wilcoxon signed-rank test, and then corrected for multiple testing with the Benjamini–Hochberg method[45]. Statistical parameters including the sample sizes, measures of distribution and $P$ value thresholds for significance are reported directly in the figures and figure legends. In order to determine if a sample was seropositive for a given combination of antigen and antibody isotype, a cut-off value (mean + 2 standard deviations) was calculated from the negative control samples. Samples above this threshold were classified as seropositive for that antigen and isotype combination. Samples that were seropositive for any of the antibodies tested were classified as such ("anySero", Fig. 2a). Where correlations are presented, the Spearman correlation coefficient and associated $P$ value were calculated. The differences in symptom reporting between cases and non-cases were calculated using Fisher's exact test, the probability of becoming symptom-free was estimated using Kaplan–Meier's method, and the hypothesis testing was performed using the log-rank test. In order to estimate the effect of patient characteristics and antibodies on symptoms over time, we constructed a mixed-effects model. For each subject and for each timepoint, we added together with the number of complaints per symptom category. We specified a mixed-effects model per symptom category with symptoms as the response and study day, age, and sex, as explanatory variables. We also added study day and Sample_ID as random effects. The formula for the model (in R notation):

$$\text{Symptom\_count} \sim \text{Study day} + \text{Age} + \text{Sex} + \text{Viral load} + (\text{Study day}|\text{Sample\_ID})$$

In order to determine the effect of antibodies on the symptom response, the model above was updated in a univariate fashion with each antibody measurement as a covariate. The formula of the updated model:

$$\text{Symptom\_count} \sim \text{Study day} + \text{Age} + \text{Sex} + \text{Viral load} + \text{Antibody} + (\text{Study day}|\text{Sample\_ID}).$$

Estimates for the covariates, as well as 95% confidence intervals and $P$ values (Satterthwaite's approximation to degrees of freedom), were extracted and plotted.

**Reporting summary**. Further information on research design is available in the Nature Research Reporting Summary linked to this article.

## Data availability

The number of COVID-19 hospitalisations in the Netherlands was derived from https://www.rivm.nl/coronavirus-covid-19/grafieken. The processed data generated in this study are provided in the Source Data file. The raw data are available from the corresponding author upon reasonable request. The raw data are not publicly available due to data an patient's privacy laws. Source data are provided with this paper.

## Code availability

The R-code that supports the findings of this study is available from the corresponding author upon reasonable request. The raw code is not publicly available due to data and patient's privacy laws.

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

## Acknowledgements

We would like to thank all MuCo study participants for their willingness to participate and their support for the study. In addition, we thank M. Boonstra, M. Blok, D. van der Giessen, E. Lenssen, E. Reuvers, M. Roek and E. Wijnhoven for assisting with performing the first home visit, H. Lyoo from Utrecht University for experimental assistance, J. Heijnen, M. Dautzenberg and A. Voss for recruiting participants from the occupational health and safety officers at Rijnstate and CWZ hospitals, and the respective medical microbiology departments involved in PCR testing of hospital employees. This study was funded by the Laboratory of Medical Immunology, Radboud University Medical Center.

## Author contributions

Conceptualisation: D.A.D. and M.I.J.; study preparation and execution: J.F., D.A.D., R.P., J.R., D.T. and M.B.; lab processing: J.F., R.P., D.A.D., K.L., K.T., E.S. and C.E.G.-J.; formal analysis: J.F. and J.G.; writing—original draft preparation: J.F., and J.G.; writing—review and editing: J.F., J.G, D.A.D., T.B., M.A.H., M.I.J. and R.P.; resources: F.J.K, B.-J.B., N.G.-B., C.D. and M.N.-F.; and supervision: D.A.D. M.I.J. and D.A.D. contributed equally to this manuscript.

## Competing interests

The authors declare no competing interests.
