## [Peer Review File · Nature Communications]

Reviewers' comments:

Reviewer #1 (Remarks to the Author):

"Elevated mucosal antibody responses against SARS-CoV-2 are correlated with lower viral load and faster decrease in systemic COVID-19 symptoms", by Dr Fröberg and colleagues (manuscript number NCB-294882).

Fröberg and colleagues performed a prospective, observational household contact study (March/April 2020). They studied health care workers with a positive PCR-test (index cases) and at least two household members (contacts) in home quarantine. One PCR test was performed at the first visit, and serum was taken on day 28 after the first visit. Nasal mucosal lining fluid (MLF) was taken at the first visit, and day 3, 6, 28 (self sampling) for index cases and day 7, 14, 28 (self sampling) for contact persons. Contact cases were defined by being positive for PCR or serum Ig or mucosal Ig. Contact cases are divided in two groups: primary, who were PCR positive at the first visit, and non-primary, who became positive and thus cases later. Symptoms were documented in a questionnaire throughout the study. For PCR testing, a PCR protocol was developed by the authors. For the antibody measurements, a fluorescent-bead-based multiplex immunoassay (MIA) was newly developed by the authors. The study followed the regulations of the Declaration of Helsinki, was registered and had ethics approval.

Overview of main findings:

Index cases only (50 cases):

- Almost all index cases (96 %) seroconverted (anti-S IgG), and serum Ig levels correlated with the mucosal Ig levels (both measured at day 28 after first visit) (Fig. 2A,B)
- In index cases, at the time point of enrolment, viral load negatively correlated with mucosal IgM (Fig. 3A)
- In index cases, mucosal Ig levels increased over time starting day 7-9 after onset of symptoms (Fig. 3B)
- In index cases, the number of respiratory symptoms (but not of systemic symptoms) positively correlated with age (Fig. 4B)
- In index cases the occurrence of mucosal IgM and IgG over time coincided with an improvement of systemic but not respiratory symptoms

Contact persons (137)

- Primary contact cases (9) were positive at enrolment, non-primary contact cases became positive later (66), 62 remained negative (Fig. 5A)
- In primary contact cases at enrolment, viral load negatively correlated with mucosal IgM (Fig. 5B)
- In both primary and non-primary contact cases mucosal Ig seemed to increase over time, but no statistical evaluation is available (Fig. 5C).
- Approx. 50 % of all contact persons became infected, independent of age (Fig. 6A)
- Of infected contact cases, approx. 20 % remained asymptomatic (Fig. 6B).
- In contact cases, the number of symptoms positively correlated with age (Fig. 6C)

Conclusions supported by the results:

- Mucosal Ig levels increase over time and correlate with serum Ig levels at the end of the study, and are associated with faster resolution of systemic symptoms.
- About 50 % of contact persons (all ages) become positive for PCR or mucosal or serum Ig, and 20% of those remained asymptomatic.
- Respiratory symptoms are more severe in older persons

The authors chose to draw the following major conclusions (title abstract and discussion):

- Elevated mucosal IgM is associated with lower viral load

- Mucosal antibodies are correlated with decreases in systemic symptoms
- Older age was associated with an increase in respiratory symptoms
- Up to 42% of household contacts, including children, developed specific mucosal antibodies.
- High transmission rates within households in which children might play an important role

As pointed out in the specific comments below, some of the conclusions of the authors are not well supported. This manuscript would benefit from focusing on questions the study was designed for, and the clear findings, and not reach beyond the evidence provided. The authors should put more effort to discuss the limitations of the study design and methods (validation of assays, limited number of index patients and family members, self sampling etc). Nevertheless, this study is an important contribution performed at the very start of the pandemic in March and April 2020 when not much knowledge and tools were available for SARS-CoV-2. The manuscript should focus more on the transmission from index to contact persons, which is the key question the study is designed for, and in this context, should also discuss in greater detail the validity and the limitations of the analytical methods.

Comments

1. The analytical methods used, PCR and antibody measurements, were newly developed in their own laboratory. Are the assays validated and compared to other established analytical methods? Can you provide information about sensitivity and specificity?
2. One of the key questions at the time the study was designed and performed, was the transmission rate of infection to other household members. AnySero in Fig. 2A provides the number 53.3%. Anti-S IgG is 43.1%. Without information about sensitivity and specificity it will be difficult to correct the percentage of transmission for the use of multiple assay readouts. Furthermore, although the seroprevalence in the population at that time is probably almost zero, health care workers with their families might have a higher background, and could be used as a negative control.
3. Contact persons might have experienced respiratory or systemic symptoms that were unrelated to SARS-CoV-2. Number of symptoms in contact cases and contact non-cases should be shown. Furthermore, number of symptoms should be analysed for different age groups, and p-values calculated. For the conclusion about children, children need to be looked at as a group, and compared to different other age groups. A correlation as in Fig. 6C does confirm an age dependent effect, but does not allow a conclusion specifically about the subgroup children, since the age dependent increase in respiratory symptoms increases with age in general. Furthermore, it may turn out that the number of children below 12 is too low to reach statistical significance.
4. The authors should provide information about the severity of the disease course of index and contact cases. They should provide a table with symptoms and the odd ratios for different symptoms for all study participants.
5. An important question is whether the viral load and the peak of symptoms are correlated with the Ig levels. The data are available, and therefore such an analysis should be added.
6. Any speculation or conclusion about transmission by children should be eliminated since the study is not designed to answer this question. In this context, asymptomatic cases of contact persons in the 0-12 year group are about the same as in the 19-49 year group (Fig. 6B), and there seems no significant difference as the authors mention in the results section ($p=0.1$).
7. Abstract: Overall, the abstract text provides not sufficient information about the study performed and the results, and it lacks clarity. The study contains a lot more data than the title and the abstract

suggest.

8. Abstract: "Mucosal antibodies play a key role in protection against SARS-CoV-2 exposure, This wording suggests that the role of mucosal IgM titers was examined. However, the work does not study the functional role of IgM during first contact with virus, but is rather descriptive.

9. Abstract: "Elevated mucosal IgM was associated with lower viral load."

This conclusion is limited to index cases (50) and primary contact cases (9) and to the single time point at enrolment (first visit). What about the other 66 non-primary contact cases? Is there also a relationship of mucosal Ig levels and viral load? The conclusion requires a longitudinal analysis of viral load and development of mucosal antibody titers, which was not performed. There was only one time point of PCR testing, an obvious limitation of the study with regard to this main claim. Furthermore, the longitudinal measurements of mucosal antibodies in contact cases (primary and non-primary) do not seem to show significant increases in titers? At least a statistical analysis is lacking.

10. Abstract: "RBD and viral spike protein-specific mucosal antibodies were correlated with decreases in systemic symptoms, while older age was associated with an increase in respiratory symptoms."

First part of sentence is supported by results on mucosal IgG and IgM but not IgA S (Fig. 4C). Notably, none of the mucosal Igs correlate with respiratory symptoms despite mucosal location of those Igs; this should be mentioned in the abstract.

Second part of sentence is not related to the first part at all (while?) (Fig. 4B). Fig. 4B does only refer to symptoms and age, and not to mucosal antibodies. Fig. 4B starts with the age of 20. Was the lower respiratory symptoms analysis only performed for 20 years and older? Or are all children grouped in the 20 group? But this might be a matter of the modeling performed. The error in the model for younger individuals seems quite high. This way, no conclusion about children younger than 12 years seems possible.

11. Abstract: "Up to 42% of household contacts developed SARS-CoV-2-specific mucosal antibodies, including children, indicating high transmission rates within households in which children might play an important role."

42% is not mentioned in the results. What means "up to 42%" in this context? Who is counted as positive? Sensitivity, specificity, which Ig subtypes?

To "including children": not surprising that children develop mucosal antibodies. But the whole sentence makes no sense: in this study, children play no role in transmission, since index cases are adults. Even if in this study children show the same infection rate, and a similar viral load, there is no evidence for higher transmission from children to adults in this work.

12. Introduction (45-46): "...children often develop asymptomatic or atypical disease which makes them prone to underdiagnosis 3,9,14"

Not supported by data of this study: Fig. 6B: The percentage of asymptomatic in 13-18 year old individuals is in the range of 50 year and older. Children below 12 show a higher percentage of asymptomatic courses. However, the number of 12 year old and younger is 16. There is no statistical analysis. The data are not sufficient to make conclusions about transmission based on asymptomatic disease and age.

13. Due to the study design chosen, the authors missed the opportunity to monitor and compare the development of serum and of mucosal antibodies. Serum was only taken at the last visit day 28 (Fig. 1C)

14. Introduction (75-76): "We observed the strongest increases in mucosal antibodies for IgM and IgG directed against S and RBD early after symptom onset". According to Fig. 3B, the mucosal antibodies

increase from day 7-9 to day 15 after onset of symptoms (so not early after symptom onset), which seems a similar time course as for serum antibodies (see lines 59-60 in introduction).

15. Introduction (78-79): "Increased RBD and S-specific mucosal antibodies correlated with decreases in systemic symptoms over the study period, while older age was associated with an increase in respiratory symptoms." Similar sentence as in the abstract, two unrelated parts, see comment 10.

16. Introduction (79-81): "Finally, we demonstrate that up to 42% of participating household contacts develop antibodies to SARS-CoV-2, including children, suggesting high transmission among household contacts." Same as in abstract (see comment 11), too general, not precise: 42% develop which antibodies? Where in the results section is this number?

17. Introduction (82-83): "Child contacts were infected at a similar rate as adult contacts with similar viral load, but developed less symptoms compared to adults. Therefore the role of children in transmission might be underestimated." There is an age dependent increase in symptoms, but without specific reference to children 12 years and younger. The number of children examined is too low to support this statement. Also see comments 10-12.

18. Results: Fig. 6A: no p-value for <12 compared to >12 years in figure or figure legend, but in results text ($p=0.78$): how was the statistical analysis performed?

19. Fig. 6B: symptomatic versus asymptomatic is determined at study start, and not monitored over time. Children might get their symptoms later. Why is symptomatic versus asymptomatic only shown for the time point of enrolment? Does the same apply to Fig. 6C? Does total number of symptoms only refer to the time point of study enrolment? According to study protocol, symptoms were monitored over time. In the results section describing Fig. 6C it seems to be the total number of all symptoms over time, please specify. "Of note, 29% of all infected household members did not have any symptoms at study start, and 19% of all infected contacts remained asymptomatic throughout the entire follow-up. (226-228)". Who are the 10 % developing symptoms later, children? Symptoms over time should be provided for each individual participating in the study, and analysed. There is only one p-value provided ($p=0.1$) which indicates that the observation that more children are asymptomatic at enrolment than adults is not significant, and thus this is not a result. Asymptomatic disease implies that they were asymptomatic at any time of the study and not only at study enrolment (see legend of Fig. 6B)?

20. The primary cases of all contact cases were only 9 of 67 according to figure legend 5B. How does this relate to Fig. 5A, very confusing.

21. (193-194): "Cases were classified first by their PCR result on enrolment, followed by seroconversion for one of the antibody isotypes against S-protein and mucosal antibody responses against S-protein at the end of the follow-up." For clarity, the exact definition of a case should be included. Which anti-SARS-CoV-2 antibodies count to define cases? And in this context, the sensitivity and the specificity for each assay should be made available, and if multiple types of Ab count for the definition of a case (serum, MLF, S, RBD, N) an appropriate correction for the use of multiple assays should be applied.

22. Discussion (242-243): "This study demonstrates the unique value of using mucosal lining fluid to assess various aspects of SARS-CoV-2 infection, ranging from pathophysiology to epidemiology." What exactly is meant by unique value? What is the new insight into pathophysiology and epidemiology? How robust is the method? How sensitive, how specific, how long-lasting the mucosal antibodies beyond 4 weeks?

23. Discussion (249-250): "A key finding of this study is that the mucosal IgA response against RBD showed the strongest relation to resolution of systemic disease symptoms." How do the authors explain that IgA against RBD is a key finding, but that IgA against S in Fig. 4C is not significant? Furthermore, why are mucosal Igs of any kind not associated with the resolution of respiratory symptoms? Furthermore, in Fig. 2A, in the serum there seems to be no IgA against RBD at all, but against S, and how does that fit to reference 33 which is cited in line 254?

24. Discussion (262-262): "Another significant finding was the negative correlation between viral load and mucosal IgM against RBD and S, measured at study start, suggestive of a neutralizing role of mucosal IgM...". The negative correlation of viral load and anti-IgM is shown for index cases and for primary contact cases but not the others who subsequently develop infection. To make this conclusion, which is the main conclusion of the manuscript and as such included in the title, a longitudinal monitoring of viral load and mucosal IgM would be required. Then the decrease of an initial viral load can be related to the appearance of increasing amounts of mucosal IgM. As it is now in the paper, no functional correlation can be claimed.

25. Discussion (289-291): "The ability to measure the mucosal antibody response over a longer period of time in an easy, non-invasive manner allowed us to identify cases among the household members that were not picked up by PCR-testing at a single timepoint only." What if PCR would have been performed at the same frequency as mucosal Ig measurements? Both PCR and mucosal Ab are performed in swabs. I don't see an advantage of mucosal Ab over PCR in terms of practical issues and validity.

26. Discussion (292-294): "When we performed the study, we were unable to collect additional samples for viral PCR due to shortages in swabs and transport medium, limiting our ability to fully study the dynamic interactions between viral infection and antibody responses, and calculate the sensitivity of the antibody measurements in diagnosing infection in comparison to PCR." Very true, a weakness in study design and a drawback of the study at least for the intended conclusions.

27. Discussion (317): "Almost half of the child cases in our study were asymptomatic". This refers to a single time point at enrolment, and parents reported for their kids (Fig. 6B). The p value is 0.1 and thus not significant. No details about the calculation. This is not a valid result, and still is used to draw a conclusion about the contribution of children to transmission, here, and in the abstract. This study is not designed to answer the question whether children contribute to transmission more than adults. In such a study, the index cases have to be children, but here index cases are adults. Thus, this study should refrain from the question about children and transmission.

28. Fig. 6C: what is meant by total number of reported symptoms: up to 150, please specify, unclear.

29. Title: "faster decrease in systemic COVID19 symptoms": this is only significant for systemic symptoms, see Fig. 4C: explain how this is calculated: Fig. 4C legend: "Longitudinal mucosal antibody measurements were added as fixed effect variables in a univariate fashion to the MEM from b)". Furthermore, this refers to index patients only. And what about serum Ig titers, do the serum Igs at day 28 correlate with disease severity and decrease of systemic or respiratory symptoms?

30. Line 217 "isotypes of S and RBD antigens (Figure 5b)." should read "Figure 5C".

31. Methods (379-380): "MLF was collected on day 0, 3, 6 and 28-33". Better: one of the days 28-33.

32. In Fig 1A: what is meant by: "Secondary outcome (exposure)" in the figure? Should this not be

similar to the secondary outcomes on the left side: symptoms, serology etc? What is meant by "estimation of SARS-CoV-2 exposure" in the figure legend?

33. Fig. 5C: Are all contact persons depicted, or only those who were defined as cases based on PCR/serum Ig/mucosal Ig? Please specify in the figure legend.

34. According to Table S2, one index case and five contact cases were only MLF-positive but not PCR- or serum-positive. The five contact cases can hardly be identified in Fig. 6C. Did this group have less symptoms compared to the others? PCR was only determined at the first visit, so later viral load can be easily missed. But it would be interesting to know whether the MLF positive and serum negative group has less symptoms.

35. In this context, which assay is more sensitive, MLF Ig or serum Ig? This is especially important if the authors propose that mucosal MLF Ig is a reliable and easy to perform test method. A correlation of both measurements has been performed for index patients (Fig. 2B) but with no conclusion about sensitivity and specificity.

Reviewer #2 (Remarks to the Author):

In this manuscript, Fröberg J. et al. evaluated virus-specific mucosal antibody responses during primary SARS-CoV-2 infection to ascertain their relationship with viral load and clinical symptoms. While higher virus-specific mucosal IgM was associated with lower viral load, higher virus-specific mucosal IgM, IgG and IgA correlated with decreased systemic symptoms. Finally, a large fraction of household, including children, was found to develop virus-specific mucosal antibodies, which raises the possibility that children play a significant role in viral transmission within households.

GENERAL COMMENT

This is a descriptive but novel and timely work that deals with a quite understudied topic: the role of mucosal antibody responses in SARS-CoV-2 infection. Aside from the analysis of mucosal antibodies from households, additional strengths of this study include its prospective nature, appropriate patient cohort, data from both adults and children, and appropriate statistical analysis. In general, the main conclusions rely on convincing data. A general weakness relates to the provision of an insufficient background on antibody responses in the respiratory tract. A more specific weakness relates to the lack of more extensive data on virus-specific mucosal antibodies, including neutralizing activity data. However, authors discuss some of these weaknesses in the text. In general, authors should be praised for performing a quite systematic work under very difficult conditions. The following specific comments are provided to enhance the impact of the manuscript.

SPECIFIC COMMENTS

1) RBD is a key domain of the S protein from SARS-CoV-2. Why serum IgA to the S antigen positively correlates with seroconversion, whereas serum IgA to RBD do not (Figure 2a)? The opposite is true for serum virus-specific IgM, although differences are not as striking as those observed in serum virus-specific IgA. Some discussion may be needed.

2) I wonder about the nature of serum virus-specific IgA. What is the proportion of monomeric (= systemically produced) vs dimeric (= mucosa-associated) IgA?

3) Is there any virus-specific IgA in the stool from infected patients? This finding would suggest the involvement of the intestinal mucosa in the infection. Should fecal samples be unavailable, authors should at least discuss this important aspect of SARS-CoV-2 infection in light of the current literature.

4) Authors should strive to place their findings in a mucosal immunology context readily understandable by a broad readership. Their work specifically analyzes mucosal antibodies during SARS-CoV-2 infection. Different antibody classes are studied both in respiratory secretions and serum. What are the main differences between serum and mucosal IgM, IgA and IgG, if any? What are the predominant antibodies produced by the respiratory mucosa, under steady state or post-infection conditions? A minimal background would be really helpful.

5) I would encourage authors to replace current general, i.e., open-ended titles from both Results and Legends to Figures with more specific, i.e., explicative titles to improve the readability and overall impact of this manuscript.

Reviewer #3 (Remarks to the Author):

Froberg et al report analysis of a prospective observational SARS-Cov-2 household contact study. The study offers a number of basic insights into the relationships between infection, viral load, symptoms, and mucosal antibody responses. The household contact design offers a setting distinct from other reports that have profiled antibody responses.

The most significant observations reported are the relationship between mucosal Ig responses and decreased viral load (Fig 3a, 5b) and faster resolution of symptoms (Fig 4c). Unfortunately, the data presentation with respect to these observations seems as though it would benefit from revision.

Specifically, the correlative relationships are simply indicated with a color denoting correlation coefficient, and significance star(s). There are not too many relationships shown to present this data in raw form, providing the underlying scatterplots. The critical analysis reporting faster resolution comes in the form of "predicted change in symptoms" resulting from a model whose performance is not shown in the main figures, and which at least this reviewer had some difficulty in definitively confirming was the precise model shown in supplemental.

Given these data support the title of the manuscript, the authors would do well to better focus and bolster the data presentation on these specific points. The paper would also benefit from further reporting on the 5 MLF positive subjects that never sero-converted or tested PCR positive. Establishing naso-conversion in the absence of seroconversion – even appropriately caveated due to low subject numbers is worth noting and presenting more meaningfully (ie: with raw data).

In terms of content, discussion of evidence from animal models and other human CoV in terms of mucosal immunity would round out the abstract/intro/discussion. Similarly, it seems worthwhile to specifically highlight the importance of the observations made here in terms of their linkage to vaccines. Mucosal vaccines have yet to be developed but may play a critical role in the goal of reducing transmission.

Minor points:

"Mucosal antibodies play a key role in protection against SARS-CoV-2 exposure" – This opening sentence is confusing. I think it may be at odds with the traditional definition of "exposure".

The low IgA seroconversion for RBD and N is at odds with other reports. The authors should comment.

There is a typo in Figure 1b in the word participants. A few elsewhere as well.

What are the large dots in Fig 3b? What results in a relative ratio of zero?

It took this reviewer a long time to find the antigen labels in figure 3c. Having complete feature names on one side of the plot would be helpful. The line of identity looks like it has a correlation of 0 when of course it really has a coefficient of 1.

Fig 4 B – please plot some raw data, not just the model. Please report the model coefficient weights.

Plots like Fig 3b and 5c would benefit from column and row labels as opposed to use of individual chart titles.

I think I am missing the purpose of Fig 5c. It seems to basically show that primary cases responded faster to their own start of symptoms than household contacts did. It seems fairly obvious that a primary case would mount an earlier response than a subsequently infected contact. Is this figure meant just to validate the identification of who was primary? Is it meant to show something else – focusing on the lower magnitudes that are briefly mentioned but not explained? It seems surprising that the not primary cases do appear to have lower magnitude responses – the authors should comment on the potential basis and significance of this observation. It appears to be referred to as Fig 5b in the manuscript.

Figure S4b is too small to meaningfully view.

Manuscript review comments Nature communications

Reviewer 1 point #1. The analytical methods used, PCR and antibody measurements, were newly developed in their own laboratory. Are the assays validated and compared to other established analytical methods? Can you provide information about sensitivity and specificity?

Author response: The Luminex-based multiplex immune assay (MIA) we used in this study has indeed been established and validated, see: den Hartog G, Schepp RM, Kuijer M, GeurtsvanKessel C, van Beek J, Rots N, et al. SARS-CoV-2–Specific Antibody Detection for Seroepidemiology: A Multiplex Analysis Approach Accounting for Accurate Seroprevalence. *The Journal of Infectious Diseases*. 2020;222(9):1452-61.

We have included a ROC analysis to demonstrate the sensitivity and specificity of the MIA, which we have now also added as Figure S1a. This is also described in the results section, lines 103-107. Finally, we have also assessed the reproducibility by measuring the same sample twice more than nine months apart. This has been added as a supplementary figure (Figure S1b) and is also described in the result section, lines 108-109.

The PCR protocol and assay we used has been validated extensively (Corman VM, Landt O, Kaiser M, Molenkamp R, Meijer A, Chu DK, et al. Detection of 2019 novel coronavirus (2019-nCoV) by real-time RT-PCR. *Eurosurveillance*. 2020;25(3):2000045). We have included additional information on this assay in the methods section, see lines 347-351.

Reviewer 1 point #2. One of the key questions at the time the study was designed and performed, was the transmission rate of infection to other household members. AnySero in Fig. 2A provides the number 53.3%. Anti-S IgG is 43.1%. Without information about sensitivity and specificity it will be difficult to correct the percentage of transmission for the use of multiple assay readouts. Furthermore, although the seroprevalence in the population at that time is probably almost zero, health care workers with their families might have a higher background, and could be used as a negative control.

Author response: Because we started this study at the very beginning of the first wave in the Netherlands, we believe that the seroprevalence of anti-S antibodies at that time was close to zero (see **Figure 1a**). As demonstrated by the ROC analysis (see point 1 above), the MIA assay and the seropositivity definition is robust. We have adapted the Results section as follows, lines 110-111. We also clarify the case definition, see Methods lines 413-421. Although we believe that most - if not all - of SARS-CoV-2 transmission occurred within the household, we agree with the reviewer that the study design was not formally designed to address this question and we have therefore removed any reference to potential transmission within the household.

Reviewer 1 point #3. Contact persons might have experienced respiratory or systemic symptoms that were unrelated to SARS-CoV-2. Number of symptoms in contact cases and contact non-cases should be shown. Furthermore, number of symptoms should be analysed for different age groups, and p-values calculated. For the conclusion about children, children need to be looked at as a group, and compared to different other age groups. A correlation as in Fig. 6C does confirm an age dependent effect, but does not allow a conclusion specifically about the subgroup children, since the age dependent increase in respiratory symptoms increases with age in general. Furthermore, it may turn out that the number of children below 12 is too low to reach statistical significance.

Author response: We agree with the reviewer that some of the symptoms may be unrelated to SARS-CoV-2. To show variation in reported symptoms between cases and non-cases, we have added Figure S5a. Additionally, we have included the symptom analysis in the Results

section, lines 178-184. Finally, we have expanded the initial symptom analysis to also include the contact cases. Figure 4a-b shows the symptom development in cases stratified per age group, as well as the general influence of age on symptom development. This has also been described in the result section, lines 186-210. We have removed any claims about asymptomatic children and their role in transmission, as the reviewer correctly pointed out that the study was not adequately powered to address this question.

Reviewer 1 point #4. The authors should provide information about the severity of the disease course of index and contact cases. They should provide a table with symptoms and the odd ratios for different symptoms for all study participants.

Author response: Please see our response to point 3 above. We have amended the manuscript to include a figure with the reported symptoms per age group for cases and non-cases, ordered by the difference in reporting between the cases and non-cases: Figure S5a. The bolded symptoms in this figure were more often reported in the COVID-19 cases, as defined by a Fisher's exact test. This figure provides insight into which symptoms are more COVID-19 specific, and which age groups reported the most symptoms.

Reviewer 1 point #5. An important question is whether the viral load and the peak of symptoms are correlated with the Ig levels. The data are available, and therefore such an analysis should be added.

Author response: We agree with the reviewer and have therefore included the day 0 viral load in the mixed effect model to correct for the effect of viral load on the symptom development and effect of mucosal antibodies. Viral load did not have an influence on the symptom progression, which has been stated in the result section, lines 201-202. Since we only determined viral load at day 0, we were unable to directly relate the viral load to symptom resolution.

Reviewer 1 point #6. Any speculation or conclusion about transmission by children should be eliminated since the study is not designed to answer this question. In this context, asymptomatic cases of contact persons in the 0-12 year group are about the same as in the 19-49 year group (Fig. 6B), and there seems no significant difference as the authors mention in the results section ($p=0.1$).

Author response: We thank the reviewer for the critical comments on the role of children in viral transmission. Although we believe that there is increasing evidence that children play a larger than previously assumed role in viral transmission already during the first wave, we also fully acknowledge that our study was not designed to address this particular question. We have therefore removed any claim about transmission by children from the manuscript. Instead, we have focused our symptom analyses on three different age groups and the effect of age on clinical symptoms in general, see point #3.

Reviewer 1 point #7. Abstract: Overall, the abstract text provides not sufficient information about the study performed and the results, and it lacks clarity. The study contains a lot more data than the title and the abstract suggest.

Author response: We have significantly modified the abstract to provide a more complete summary of the study results, while keeping to the 100 word-count limit.

Reviewer 1 point #8. Abstract: "Mucosal antibodies play a key role in protection against SARS-CoV-2 exposure, This wording suggests that the role of mucosal IgM titers was examined. However, the work does not study the functional role of IgM during first contact with virus, but is rather descriptive.

Author response: This sentence was based on a general consensus that mucosal antibodies are important in the protection against re-infection, and our finding that IgM may play an important role in this process. We acknowledge that at this time no conclusive evidence is available yet on the clinical significance of mucosal IgM. We have therefore amended the abstract, see lines 26-28 in the abstract.

Reviewer 1 point #9. Abstract: "Elevated mucosal IgM was associated with lower viral load."

This conclusion is limited to index cases (50) and primary contact cases (9) and to the single time point at enrolment (first visit). What about the other 66 non-primary contact cases? Is there also a relationship of mucosal Ig levels and viral load? The conclusion requires a longitudinal analysis of viral load and development of mucosal antibody titers, which was not performed. There was only one time point of PCR testing, an obvious limitation of the study with regard to this main claim. Furthermore, the longitudinal measurements of mucosal antibodies in contact cases (primary and non-primary) do not seem to show significant increases in titers? At least a statistical analysis is lacking.

Author response: Based on the reviewer's suggestion we have changed the analysis to include all PCR+ cases. This has been more clearly described in the figure legends of Figure 3 and Figure S4, and has been visualized in the flow-chart of Figure 1a. The fact that we only have a single PCR measurement is indeed a limitation of our study, as also acknowledged in the discussion, but should also be viewed in the context of the time of the pandemic when this study was performed, i.e. at a time when there was a national shortage of swabs and PCR reagents. This has been mentioned in the discussion section, lines 268-271.

Reviewer 1 point #10. Abstract: "RBD and viral spike protein-specific mucosal antibodies were correlated with decreases in systemic symptoms, while older age was associated with an increase in respiratory symptoms." First part of sentence is supported by results on mucosal IgG and IgM but not IgA S (Fig. 4C). Notably, none of the mucosal Igs correlate with respiratory symptoms despite mucosal location of those Igs; this should be mentioned in the abstract. Second part of sentence is not related to the first part at all (while?) (Fig. 4B). Fig. 4B does only refer to symptoms and age, and not to mucosal antibodies. Fig. 4B starts with the age of 20. Was the lower respiratory symptoms analysis only performed for 20 years and older? Or are all children grouped in the 20 group? But this might be a matter of the modeling performed. The error in the model for younger individuals seems quite high. This way, no conclusion about children younger than 12 years seems possible.

Author response: In our previous analysis, the mixed effects modelling was only performed on the index cases, which explains why children were not included in Fig 4. We have now included all cases in the analysis, including children, and we find a significant effect of age on the number of reported COVID-19 symptoms. The abstract has been changed accordingly, see lines 31-32. Initially, we analysed the effect of age on symptom development without accounting for the effect of mucosal antibodies (see Figure 4b and point #3). However, the effect of age was also taken into account in the analysis of mucosal antibodies, where it remained significant in all cases. This has been added to the result section, lines 202-210.

Reviewer 1 point #11. Abstract: "Up to 42% of household contacts developed SARS-CoV-2-specific mucosal antibodies, including children, indicating high transmission rates within households in which children might play an important role." 42% is not mentioned in the results. What means "up to 42%" in this context? Who is counted as positive? Sensitivity, specificity, which Ig subtypes? To "including children": not surprising that children develop mucosal antibodies. But the whole sentence makes no sense: in this study, children play no

role in transmission, since index cases are adults. Even if in this study children show the same infection rate, and a similar viral load, there is no evidence for higher transmission from children to adults in this work.

Author response: The 42% refers to the proportion of household members who developed a mucosal IgA, IgM or IgG response against S, which was indeed not mentioned in the Results section. For clarity, we have removed this sentence from the abstract. Cases are defined based on either PCR and/or seropositivity, see also points #1 and #2 above.

Because of the testing policy in the Netherlands at the time of the study, only healthcare workers were eligible for PCR testing. Consequently, index cases were by definition adults. However, an index case is not necessarily the primary case in a household and it remains very well possible that children were in fact the primary source of infection in the household. In general, because infected children are less symptomatic than older infected cases and the starting point for PCR is generally symptom development, the odds of children being identified as the primary case are significantly reduced. This is now increasingly being recognized and many countries have implemented weekly rapid antigen self-testing in school-settings to identify asymptomatic infections. Nonetheless, as already discussed extensively above, our study design was not properly designed to address this question and we have therefore omitted any reference to children .

Reviewer 1 point #12. Introduction (45-46): "...children often develop asymptomatic or atypical disease which makes them prone to underdiagnosis 3,9,14"

Not supported by data of this study: Fig. 6B: The percentage of asymptomatic in 13-18 year old individuals is in the range of 50 year and older. Children below 12 show a higher percentage of asymptomatic courses. However, the number of 12 year old and younger is 16. There is no statistical analysis. The data are not sufficient to make conclusions about transmission based on asymptomatic disease and age.

Author response: This sentence was removed from the introduction since we decided to put less emphasis on the children in our study. As such, we now extended the group to include all children aged 0-18y (n=51). Less symptoms were observed in this age group compared to the older age groups (Figure 4a and S4a), in line with many other observations, see Chang MC, Park Y-K, Kim B-O, Park D. Risk factors for disease progression in COVID-19 patients. BMC Infectious Diseases. 2020;20(1):445. & Ludvigsson, J. F. (2020). Systematic review of COVID-19 in children shows milder cases and a better prognosis than adults. Acta paediatrica, 109(6), 1088-1095. As the reviewer pointed out, it is possible that this is mostly driven by children under the age of 12, but the study is not adequately powered to perform such subgroup analyses.

Reviewer 1 point #13. Due to the study design chosen, the authors missed the opportunity to monitor and compare the development of serum and of mucosal antibodies. Serum was only taken at the last visit day 28 (Fig. 1C)

Author response: We agree with the reviewer that this was a missed opportunity. We initiated this study in a very short period of time because we wanted to capture the first wave. Although we initially planned this study as a mucosal study only, we realized the importance of possibly comparisons with serum antibodies. Following an ethics amendment, we sampled serum at day 28 and nine months. This limitation has been added to the discussion section, lines 271-274. Although we agree that this is a limitation of the study, we can still compare post-infection antibody responses in serum and MLF and we believe that the inclusion of multiple mucosal antibody sampling time points represents relevant new information.

Reviewer 1 point #14. Introduction (75-76): "We observed the strongest increases in

mucosal antibodies for IgM and IgG directed against S and RBD early after symptom onset". According to Fig. 3B, the mucosal antibodies increase from day 7-9 to day 15 after onset of symptoms (so not early after symptom onset), which seems a similar time course as for serum antibodies (see lines 59-60 in introduction).

Author response: Mucosal IgM and IgG against S/RBD are significantly higher at 7-9 days after symptom onset. Serum antibodies are generally detectable approximately 10-14 days after the first day of symptoms, see also Wölfel R, Corman VM, Guggemos W, Seilmaier M, Zange S, Müller MA, et al. Virological assessment of hospitalized patients with COVID-2019. Nature. 2020, and Zhao J, Yuan Q, Wang H, Liu W, Liao X, Su Y, et al. Antibody responses to SARS-CoV-2 in patients of novel coronavirus disease 2019. Clin Infect Dis. 2020.

Since we did not collect serum at this time point, we cannot make any claim as to whether nasal antibodies are produced earlier or not than serum antibodies. Serum antibodies remain the gold standard. We make no claim about measurement of mucosal antibodies being more sensitive or 'better' than serum antibodies. However, since the respiratory tract is the primary site of infection, better insight into the mucosal antibody response is highly relevant, as it is the site where antibodies can have a direct protective effect. There are also other advantages related to the non-invasive nature that might make mucosal antibody analysis attractive and as such we see MLF sampling as highly complementary to e.g. seroprevalence studies.

Reviewer 1 point #15. Introduction (78-79): "Increased RBD and S-specific mucosal antibodies correlated with decreases in systemic symptoms over the study period, while older age was associated with an increase in respiratory symptoms." Similar sentence as in the abstract, two unrelated parts, see comment 10.

Author response: We have modified this sentence in the introduction, see lines 71-73.

Reviewer 1 point #16. Introduction (79-81): "Finally, we demonstrate that up to 42% of participating household contacts develop antibodies to SARS-CoV-2, including children, suggesting high transmission among household contacts." Same as in abstract (see comment 11), too general, not precise: 42% develop which antibodies? Where in the results section is this number?

Author response: Please see point #11 above.

Reviewer 1 point #17. Introduction (82-83): "Child contacts were infected at a similar rate as adult contacts with similar viral load, but developed less symptoms compared to adults. Therefore the role of children in transmission might be underestimated." There is an age dependent increase in symptoms, but without specific reference to children 12 years and younger. The number of children examined is too low to support this statement. Also see comments 10-12.

Author response: This is in line with many of the reviewer's previous comments (points #6, #10, #11, and #12). We have reduced the number of age groups to represent broader age categories (see point #3). Consequently, we have removed this sentence from the introduction.

Reviewer 1 point #18. Results: Fig. 6A: no p-value for <12 compared to >12 years in figure or figure legend, but in results text (p=0.78): how was the statistical analysis performed?

Author response: Analysis was performed using a Fisher's exact test. However, we have removed this analysis from the manuscript now. Details on all analyses are described in the methods section and in the figure legends.

Reviewer 1 point #19. Fig. 6B: symptomatic versus asymptomatic is determined at study start, and not monitored over time. Children might get their symptoms later. Why is symptomatic versus asymptomatic only shown for the time point of enrolment? Does the same apply to Fig. 6C? Does total number of symptoms only refer to the time point of study enrolment? According to study protocol, symptoms were monitored over time. In the results section describing Fig. 6C it seems to be the total number of all symptoms over time, please specify. "Of note, 29% of all infected household members did not have any symptoms at study start, and 19% of all infected contacts remained asymptomatic throughout the entire follow-up. (226-228)". Who are the 10 % developing symptoms later, children? Symptoms over time should be provided for each individual participating in the study, and analysed. There is only one p-value provided ($p=0.1$) which indicates that the observation that more children are asymptomatic at enrolment than adults is not significant, and thus this is not a result.

Asymptomatic disease implies that they were asymptomatic at any time of the study and not only at study enrolment (see legend of Fig. 6B)?

Author response: In line with the reviewer's earlier questions about the role of children, we have removed this part of the analysis and instead focused on the relation between symptom development within the three age groups and mucosal antibodies, see point #3. This analysis is based on all reported symptoms during the total study period from inclusion at day 0 until 28 days later. Individual symptom data is shown in Figure S6.

Reviewer 1 point #20. The primary cases of all contact cases were only 9 of 67 according to figure legend 5B. How does this relate to Fig. 5A, very confusing.

Author response: We agree that the identification of primary cases may not have been very clear. Given the difficulties and remaining uncertainty in the identification of primary cases in each household, this analysis was removed from this manuscript.

Reviewer 1 point #21. (193-194): "Cases were classified first by their PCR result on enrolment, followed by seroconversion for one of the antibody isotypes against S-protein and mucosal antibody responses against S-protein at the end of the follow-up." For clarity, the exact definition of a case should be included. Which anti-SARS-CoV-2 antibodies count to define cases? And in this context, the sensitivity and the specificity for each assay should be made available, and if multiple types of Ab count for the definition of a case (serum, MLF, S, RBD, N) an appropriate correction for the use of multiple assays should be applied.

Author response: The case definition is included in the Methods section, see also point #2. Additionally, a ROC analysis has been included for the MIA to demonstrate the performance of this assay, see point #1. We used serum IgG, IgM or IgA antibody levels against the spike protein at day 28 to identify seropositive contacts (i.e. contact cases).

Reviewer 1 point #22. Discussion (242-243): "This study demonstrates the unique value of using mucosal lining fluid to assess various aspects of SARS-CoV-2 infection, ranging from pathophysiology to epidemiology." What exactly is meant by unique value? What is the new insight into pathophysiology and epidemiology? How robust is the method? How sensitive, how specific, how long-lasting the mucosal antibodies beyond 4 weeks?

Author response: We have included novel data on the mucosal and serum antibody levels of the cases at nine months after the study start, which clearly demonstrates the persistence of mucosal (and serum) antibodies (Figures 2b, 2c and S3). Moreover, we find potentially relevant differences in persistence between mucosal and serum antibodies, suggesting that mucosal responses may provide additional information on the immune response against

SARS-CoV-2. Importantly, this study seeks to contribute new knowledge about mucosal antibodies in the respiratory tract, which has thus far been largely understudied. Regarding the sensitivity and specificity, we have included ROC analyses to demonstrate the robustness of the method, see point #1. Finally, as may be expected some of the findings derived from analysing the mucosal lining fluid samples correlate well with results obtained from serum. We are not suggesting in any way that MLF should replace serology as the gold standard. Nonetheless, whilst fingerprick blood can also be self-sampled by participants, it is far easier to ask participants to self-sample mucosal lining fluid, which can be done without any issues even from very young children and may even be repeated daily. As such, we strongly believe that mucosal lining fluid – especially when applied more extensively – does provide unique value.

Reviewer 1 point #23. Discussion (249-250): "A key finding of this study is that the mucosal IgA response against RBD showed the strongest relation to resolution of systemic disease symptoms." How do the authors explain that IgA against RBD is a key finding, but that IgA against S in Fig. 4C is not significant? Furthermore, why are mucosal Igs of any kind not associated with the resolution of respiratory symptoms? Furthermore, in Fig. 2A, in the serum there seems to be no IgA against RBD at all, but against S, and how does that fit to reference 33 which is cited in line 254?

Author response: Author response: To present the data in a more structured manner, we re-analyzed all data on nasal antibody responses, including the contact cases. Following inclusion of the contact cases, IgA-S is now also found to be significantly associated with reduction in systemic symptoms. A potential reason for the stronger effect of anti-RBD IgA as compared to the spike protein could be because antibodies against epitopes in the RBD region are the most important for neutralization of viral binding to the ACE2 cellular receptor.

As the reviewer pointed out, we did not find a relation between respiratory symptom reduction and any of the mucosal antibodies. This could partly be due to respiratory symptoms being reported much more often than systemic symptoms, both by cases as well as by non-cases. As such, RS are less COVID-19 specific than SDS (**Figure S6a**). Another potential explanation may be differences in the onset and kinetics of respiratory vs systemic disease symptoms in relation to the production of mucosal antibodies. This is also included in the discussion, see lines 244-252.

Reviewer 1 point #24. Discussion (262-262): "Another significant finding was the negative correlation between viral load and mucosal IgM against RBD and S, measured at study start, suggestive of a neutralizing role of mucosal IgM...". The negative correlation of viral load and anti-IgM is shown for index cases and for primary contact cases but not the others who subsequently develop infection. To make this conclusion, which is the main conclusion of the manuscript and as such included in the title, a longitudinal monitoring of viral load and mucosal IgM would be required. Then the decrease of an initial viral load can be related to the appearance of increasing amounts of mucosal IgM. As it is now in the paper, no functional correlation can be claimed.

Author response: Although longitudinal paired analysis of viral load and mucosal IgM would indeed provide more compelling evidence with regards to a role for IgM in viral clearance, we disagree that no conclusions can be drawn at all based on analysis from a single time point. Since the day 0 measurement captures variation between study subjects with regards to time of infection, i.e. some subjects were included later after the first day of symptoms than others, the day 0 time point includes relevant biological variation both in IgM

levels and viral load. As mentioned above in points #9 and #20, we have now performed the correlation analyses including all PCR positive cases.

Reviewer 1 point #25. Discussion (289-291): "The ability to measure the mucosal antibody response over a longer period of time in an easy, non-invasive manner allowed us to identify cases among the household members that were not picked up by PCR-testing at a single timepoint only." What if PCR would have been performed at the same frequency as mucosal Ig measurements? Both PCR and mucosal Ab are performed in swabs. I don't see an advantage of mucosal Ab over PCR in terms of practical issues and validity.

Author response: The primary purpose of PCR swabs is direct detection of potential pathogens. Although PCR swabs can also be used to detect antibodies, it is far less standardized than MLF. MLF has been designed as a precision sampling technique to detect soluble mediators (REF). Additionally, PCR swabs are much more invasive compared to MLF collection, as the PCR swabs are designed to sample the nasopharynx and generally generate more discomfort. In contrast, the nasosorption devices we use sample nasal lining fluid from the inferior turbinate, and are very well-tolerated. There are obviously pros and cons with both sampling methods regarding reproducibility, sensitivity and tolerability. However, we think it is beyond the scope of this manuscript to directly compare these methods and believe that the mucosal antibody responses should be evaluated as having merit on their own. To illustrate the MLF sampling method, we have included Figure S7. This has also been mentioned in the discussion, see also point #9.

Reviewer 1 point #26. Discussion (292-294): "When we performed the study, we were unable to collect additional samples for viral PCR due to shortages in swabs and transport medium, limiting our ability to fully study the dynamic interactions between viral infection and antibody responses, and calculate the sensitivity of the antibody measurements in diagnosing infection in comparison to PCR." Very true, a weakness in study design and a drawback of the study at least for the intended conclusions.

Author response: Yes, and already clearly mentioned as a limitation in the discussion. Although a dynamic assessment of viral load would have been ideal and the lack of such data may prevent us from drawing certain conclusions, the study was conducted during the first pandemic wave with limited resources and reagents. We also believe the lack of multiple PCR samples does not invalidate our results and conclusions. We have significantly amended the analyses and the manuscript and strongly believe that the conclusions presented in this version are supported by data.

Reviewer 1 point #27. Discussion (317): "Almost half of the child cases in our study were asymptomatic". This refers to a single time point at enrolment, and parents reported for their kids (Fig. 6B). The p value is 0.1 and thus not significant. No details about the calculation. This is not a valid result, and still is used to draw a conclusion about the contribution of children to transmission, here, and in the abstract. This study is not designed to answer the question whether children contribute to transmission more than adults. In such a study, the index cases have to be children, but here index cases are adults. Thus, this study should refrain from the question about children and transmission.

Author response: The rationale for undertaking this analysis at the time was that the starting point of PCR testing is having symptoms. Since children present with less or no symptoms during infection, they are therefore much less likely to be tested by PCR. However, we do agree that the study was not formally designed to examine transmission from children and as such we have removed all statements from the manuscript regarding this aspect. For more information, see also point #6.

Reviewer 1 point #28. Fig. 6C: what is meant by total number of reported symptoms: up to 150, please specify, unclear.

Author response: Symptoms were recorded per day, so the total number of symptoms is the cumulative number of reported symptoms per day during the complete follow-up between day 0-28 . Please note that this figure is not part of the current manuscript version anymore.

Reviewer 1 point #29. Title: "faster decrease in systemic COVID19 symptoms": this is only significant for systemic symptoms, see Fig. 4C: explain how this is calculated: Fig. 4C legend: "Longitudinal mucosal antibody measurements were added as fixed effect variables in a univariate fashion to the MEM from b)". Furthermore, this refers to index patients only. And what about serum Ig titers, do the serum Igs at day 28 correlate with disease severity and decrease of systemic or respiratory symptoms?

Author response: We have included more information about the model we used to generate the results, see figure 4 and the Methods section. The analysis now includes all cases instead of only the index cases. Serum antibody measurements were not included in this analysis, because the mixed effects model assesses changes in mucosal antibodies over time and this data is missing for the serum antibody levels.

Reviewer 1 point #30. Line 217 "isotypes of S and RBD antigens (Figure 5b)." should read "Figure 5C".

Author response: Figure references in the text have been corrected.

Reviewer 1 point #31. Methods (379-380): "MLF was collected on day 0, 3, 6 and 28-33". Better: one of the days 28-33.

Author response: This sentence is changed accordingly, see lines 322-325 .

Reviewer 1 point #32. In Fig 1A: what is meant by: "Secondary outcome (exposure)" in the figure? Should this not be similar to the secondary outcomes on the left side: symptoms, serology etc? What is meant by "estimation of SARS-CoV-2 exposure" in the figure legend?

Author response: Figure 1a has been updated to clarify the analysis strategy for this study.

Reviewer 1 point #33. Fig. 5C: Are all contact persons depicted, or only those who were defined as cases based on PCR/serum Ig/mucosal Ig? Please specify in the figure legend.

Author response: This analysis was based on cases only, see also the Figure legend. Please note that this figure is not part of the current manuscript anymore, as we have now analysed all household members together with the index cases, see also point #3.

Reviewer 1 point #34. According to Table S2, one index case and five contact cases were only MLF-positive but not PCR- or serum-positive. The five contact cases can hardly be identified in Fig. 6C. Did this group have less symptoms compared to the others? PCR was only determined at the first visit, so later viral load can be easily missed. But it would be interesting to know whether the MLF positive and serum negative group has less symptoms.

Author response: We did not find specific differences in the MLF+ sero- cases at the time. As already mentioned before, given that serum antibody responses remain the gold standard for antibody-based identification of cases, we removed naso-positivity from the case identification and included it as an exploratory analysis only. Almost all PCR+sero+ cases also become naso-positive for the spike protein (see tables S1 and S2).

Reviewer 1 point #35. In this context, which assay is more sensitive, MLF Ig or serum Ig?

This is especially important if the authors propose that mucosal MLF Ig is a reliable and easy to perform test method. A correlation of both measurements has been performed for index patients (Fig. 2B) but with no conclusion about sensitivity and specificity.

Author response: This study does not wish to make any claims about the use of MLF sampling for diagnostic purposes. Based on the ROC analysis, serum generally performed better than MLF in this particular setting. However, this is also partially dependent on the lower number of pre-COVID-19 MLF control samples that were available to us than serum samples, and we expect that inclusion of more MLF samples will improve sensitivity and specificity. Based on other (unpublished) clinical studies performed by our group, the predictive value of MLF is highly dependent on the situation. For instance, in populations with pre-existing immunity to respiratory pathogens, MLF was actually much better than serum in predicting protection against infection, whereas in a naïve population serum may be superior.

Reviewer 2 point #1. RBD is a key domain of the S protein from SARS-CoV-2. Why serum IgA to the S antigen positively correlates with seroconversion, whereas serum IgA to RBD do not (Figure 2a)? The opposite is true for serum virus-specific IgM, although differences are not as striking as those observed in serum virus-specific IgA. Some discussion may be needed.

Author response: See point #23 reviewer 1.

Reviewer 2 point #2. I wonder about the nature of serum virus-specific IgA. What is the proportion of monomeric (= systemically produced) vs dimeric (= mucosa-associated) IgA?

Author response: Serum IgA is generally monomeric whereas secreted IgA is predominately dimeric. This is an area of active research in our laboratory but beyond the scope of this particular study.

Reviewer 2 point #3. Is there any virus-specific IgA in the stool from infected patients? This finding would suggest the involvement of the intestinal mucosa in the infection. Should fecal samples be unavailable, authors should at least discuss this important aspect of SARS-CoV-2 infection in light of the current literature.

Author response: We did not collect stool from infected patients. Although the gut intestinal immune response to SARS-CoV-2 is very interesting, the current study is focused on nasal antibodies. As such, we do not believe the manuscript will be improved by including a discussion on this particular topic.

Reviewer 2 point #4. Authors should strive to place their findings in a mucosal immunology context readily understandable by a broad readership. Their work specifically analyzes mucosal antibodies during SARS-CoV-2 infection. Different antibody classes are studied both in respiratory secretions and serum. What are the main differences between serum and mucosal IgM, IgA and IgG, if any? What are the predominant antibodies produced by the respiratory mucos, under steady state or post-infection conditions? A minimal background would be really helpful.

Author response: We thank the reviewer for this constructive criticism and have added a more general background about the differences between mucosal and systemic antibodies in the introduction, see lines 60-63.

Reviewer 2 point #5. I would encourage authors to replace current general, i.e., open-ended titles from both Results and Legends to Figures with more specific, i.e., explicative titles to improve the readability and overall impact of this manuscript.

Author response: We have amended the subtitles of the result section and figures to improve readability.

Reviewer 3 point #1. The correlative relationships are simply indicated with a color denoting correlation coefficient, and significance star(s). There are not too many relationships shown to present this data in raw form, providing the underlying scatterplots. The critical analysis reporting faster resolution comes in the form of “predicted change in symptoms” resulting from a model whose performance is not shown in the main figures, and which at least this reviewer had some difficulty in definitively confirming was the precise model shown in supplemental.

Author response: We have added more information about the model used for the results given in figure 4 to the figure, see point #29 of reviewer 1. Figure S6 shows the model fit for each individual symptom trajectory, where the solid line is the number of predicted symptoms and the dots and faded line are the actual reported number. We have added two examples of the raw correlation plots of figure 3a to the main figure, and added the rest as supplementary figures, which can be seen in figure S4.

Reviewer 3 point #2. Given these data support the title of the manuscript, the authors would do well to better focus and bolster the data presentation on these specific points. The paper would also benefit from further reporting on the 5 MLF positive subjects that never sero-converted or tested PCR positive. Establishing naso-conversion in the absence of seroconversion – even appropriately caveated due to low subject numbers is worth noting and presenting more meaningfully (ie: with raw data).

Author response: We have now focused the data analysis more on the main conclusions of the analysis and made the analyses more robust by analysing the study cohort as a whole. We have removed the mucosal lining fluid from the case definition, but have assessed “nasoconversion” in an explorative manner. We did not find specific differences in the MLF+ sero- cases at the time. See also point #34 of reviewer 1.

Reviewer 3 point #3. In terms of content, discussion of evidence from animal models and other human CoV in terms of mucosal immunity would round out the abstract/intro/discussion. Similarly, it seems worthwhile to specifically highlight the importance of the observations made here in terms of their linkage to vaccines. Mucosal vaccines have yet to be developed but may play a critical role in the goal of reducing transmission.

Author response: We thank the reviewer for these excellent suggestions and have included additional information about other human coronaviruses in the introduction (Lines 56-60), as well as a broader discussion about the role of monitoring mucosal antibodies after vaccination (Lines 283-286).

Reviewer 3 point #4. “Mucosal antibodies play a key role in protection against SARS-CoV-2 exposure” – This opening sentence is confusing. I think it may be at odds with the traditional definition of “exposure”.

Author response: This sentence has now been changed, see also point #8 of reviewer 1.

Reviewer 3 point #5. The low IgA seroconversion for RBD and N is at odds with other reports. The authors should comment.

Author response: We have added this notion to the discussion, see also point #23 reviewer 1.

Reviewer 3 point #6. There is a typo in Figure 1b in the word participants. A few elsewhere as well.

Author response: Noted and changed

Reviewer 3 point #7. What are the large dots in Fig 3b? What results in a relative ratio of zero?

Author response: The large dots represent outliers. Previously, the samples that were below the limit of detection (LOD) had been given a value of zero, resulting in a relative ratio of zero. In the current manuscript we have corrected this and have given the samples that were <LOD the mean LOD of all the samples $-1 \times \text{sd}$. We have changed the axis-labels accordingly.

Reviewer 3 point #8. It took this reviewer a long time to find the antigen labels in figure 3c. Having complete feature names on one side of the plot would be helpful. The line of identity looks like it has a correlation of 0 when of course it really has a coefficient of 1.

Author response: We have decided to remove this figure from the manuscript as it did not contribute much to the story line.

Reviewer 3 point #10. Fig 4 B – please plot some raw data, not just the model. Please report the model coefficient weights.

Author response: As suggested by the reviewer, we have both added additional information on the MEM model to Figure xx. Raw data plots are shown for a selection of antibody response features in the main figure the other raw data plots are shown in Figure S6. See also point #29 of reviewer 1.

Reviewer 3 point #11. Plots like Fig 3b and 5c would benefit from column and row labels as opposed to use of individual chart titles.

Author response: We have changed the x and y axes labels as suggested to improve readability.

Reviewer 3 point #12. I think I am missing the purpose of Fig 5c. It seems to basically show that primary cases responded faster to their own start of symptoms than household contacts did. It seems fairly obvious that a primary case would mount an earlier response than a subsequently infected contact. Is this figure meant just to validate the identification or who was primary? Is it meant to show something else – focusing on the lower magnitudes that are briefly mentioned but not explained? It seems surprising that the not primary cases do appear to have lower magnitude responses – the authors should comment on the potential basis and significance of this observation. It appears to be referred to as Fig 5b in the manuscript.

Author response: This figure is not part of the manuscript anymore, as the household members have now been analysed together with the index cases, see also Figure 2c and point #3 of reviewer 1.

Reviewer 3 point #13. Figure S4b is too small to meaningfully view.

Author response: We apologize for this. The original Figure S4b has been removed and individual symptom developments are now shown in Figure S6.

REVIEWERS' COMMENTS

Reviewer #1 (Remarks to the Author):

The authors have carefully addressed all points raised. They added detailed information about the specifications of the assays used, including ROC curves (Fig. S2A and not S1A as in their reply), and added new data acquired 9 months after initiation of the study. The study is one of the few studies capturing the first wave of the SARS-CoV-2 pandemic.

Minor points:

1. Suggested change of title: "Development and persistence of mucosal antibodies following mild SARS-CoV-2 infection and their relation to viral load and COVID-19 symptoms"
2. Abstract: remain instead of remains?
3. Abstract: "...were followed up for nine months" better: "...were analysed for up to nine months"
4. Affiliations are partially written in small letters, correct?

Reviewer #3 (Remarks to the Author):

The authors have thoughtfully responded to a wide array of reviewer suggestions. The observation of the relationship between mucosal responses and outcomes is of high importance. The manuscript has been substantially improved.

Manuscript review comments Nature communications

Reviewer 1 point #1. Suggested change of title: "Development and persistence of mucosal antibodies following mild SARS-CoV-2 infection and their relation to viral load and COVID-19 symptoms"

Author response: The title has been changed to conform to the word limit of Nature communications, so this adaption is not possible anymore

Reviewer 1 point #2. Abstract: remain instead of remains?

Author response: Yes, this is correct. We thank the reviewer and have changed this in the abstract.

Reviewer 1 point #3. Abstract: "...were followed up for nine months" better: "...were analysed for up to nine months"

Author response: We agree with this change, and have changed the sentence in the abstract

Reviewer 1 point #4. Affiliations are partially written in small letters, correct?

Author response: Thank you for this remark, we have changed the affiliations so that they are all in the same style.